

# Optimal selection of satellite XCO2 images over cities for urban CO2 emission monitoring using a global adaptive-mesh model

Alexandre Danjou[1], Grégoire Broquet[1], Andrew Schuh[2], François-Marie Bréon[1], and Thomas Lauvaux[1,3]

[1]Laboratoire des Sciences du Climat et de l'Environnement (LSCE), IPSL, CEA-CNRS-UVSQ, 91191 Gif sur Yvette, France
[2]Cooperative Institute for Research in the Atmosphere (CIRA), Colorado State University, Fort Collins, USA
[3]Molecular and Atmospheric Spectrometry Group (GSMA) – UMR 7331, University of Reims Champagne Ardenne, 51687 Reims, France

**Correspondence:** Alexandre Danjou (alexandre.danjou@lsce.ipsl.fr)

**Abstract.** There is a growing interest in estimating urban $CO_2$ emission from space-borne imagery of $XCO_2$. Emission estimation methods are already being tested and applied to actual or synthetic images. However, we still need automatic and standard methods, as well as objective criteria for selecting the images to be processed. This study shows the performance of an automated process for estimating urban emissions, standardised for all cities, using synthetic satellite images of $XCO_2$. We

also use a decision tree learning method to define satellite image selection criteria.

We show that our method, based on a Gaussian plume model, has a success rate of 92% when applied to our database of 9920 images covering 31 cities worldwide. Using our learning method, we show that the two main criteria guiding the error on the emission estimate are the wind direction's spatial variability and the targeted city's emission budget. Our learning method also allows us to separate images giving statistically accurate estimations from those giving erroneous estimations based on

the two abovementioned criteria. Images for which the spatial variability of wind direction is low (less than 12°) and urban emissions high (greater than 12.1ktCO$_2$/h) account for 47% of images and have a bias on the emission estimation of -7% of the emissions and a spread (IQR) of 56%. Images with high spatial variability in wind direction or low urban emissions account for 53% of the images and have a bias in the emission estimate of -31% and a spread of 99%.

## 1 Introduction

Many of the most emitting countries report their $CO_2$ emissions to the UNFCC annually (19, 2013). However, despite this monitoring of emissions and the commitments made by nations to reduce them, the increase in $CO_2$ emissions continues year after year (Friedlingstein et al., 2022). Many cities worldwide have committed to reducing their emissions at their level, notably through joint initiatives such as the Covenant of Mayors (https://www.globalcovenantofmayors.org/) or the C40 cities

(https://www.c40.org/). These cities carry out self-reported inventories (SRI) based on economic data to verify the effective





reduction of their emissions. Gurney et al. (2021) compared SRIs of American cities to the Vulcan inventory (Gurney et al., 2020). This comparison shows large differences between the two datasets and highlighted the inaccuracy of the emissions estimates in most of the SRIs. Quantifying city emissions using observations of $CO_2$ concentrations above cities with satellites could provide helpful information to evaluate these assessments.

Observations from OCO-2 and OCO-3 of $CO_2$ column-average dry air mole fraction ($XCO_2$) at the scale of a few square kilometers have paved the way for quantifying emissions from large (a few $ktCO_2$/h) industrial (Chevallier et al., 2022; Nassar et al., 2017; Zheng et al., 2019) and urban (Lei et al., 2021; Reuter et al., 2019; Wu et al., 2018; Ye et al., 2020) sources of $CO_2$. Indeed, the accuracy of the observations (less than one ppm, Worden et al. (2017); Taylor et al. (2020)) is of the same order of magnitude as the XCO2 enhancements of the plumes from these sources, and their fine resolution ($\approx 2 \times 2km$, 30   Eldering et al. (2017, 2019)) allows them to capture detailed transects or images of the plumes. The Snapshot Area Map (SAM) mode of OCO-3 even provides "snapshot" images of about $80km \times 80km$ over the cities and thus a 2D coverage of the $XCO_2$ concentrations, contrary to OCO-2 and to the nominal mode of OCO-3, which only samples $XCO_2$ over a fine swath ($\approx 10km$). Studies have used these SAMs to evaluate transport model simulations (Kiel et al., 2021) or to calculate co-emitted species ratios (Lei et al., 2022; Wu et al., 2022). First estimates of city emissions based on these SAMs have been presented in 35   conferences. However, no systematic processing of SAMs over cities exists to estimate the corresponding urban emissions.

     Studies such as Broquet et al. (2018); Danjou et al. (20xx); Pillai et al. (2016); Kuhlmann et al. (2020) have used synthetic data to evaluate the possibility of quantifying $CO_2$ urban emissions from 2D XCO2 images, such as OCO-3 SAMs or simulated XCO2 images from the future CO2M (Sierk et al., 2021) and GOSAT-GW (https://www.nies.go.jp/soc/doc/IWGGMS-18/O/ 2-6_Hiroshi_Tanimoto.pdf) missions. The quantification relies on inverse modelling methods, some of which compare simula-40   tions from complex transport model to satellite observations to estimate emissions. However, Feng et al. (2016) and Lian et al. (2018) show that the WRF model (used by Lei et al. (2021) and Ye et al. (2020) with OCO-2 data) simulated $CO_2$ transport poorly when the wind speeds were low. Other emission estimation methods, called hereafter computationally-light methods, are based on simpler transport models (Gaussian plume, Danjou et al. (20xx); Krings et al. (2011)) or mass balances (Integrated Mass Enhancement method, Danjou et al. (20xx); Frankenberg et al. (2016); Varon et al. (2019)) or flux estimation (Cross-45   Sectional method, Danjou et al. (20xx); Kuhlmann et al. (2020); Krings et al. (2011); Varon et al. (2019, 2020)). Danjou et al. (20xx) evaluated these methods and again showed that attempts to quantify emissions in low wind conditions gave erroneous results. However, no established procedures exist to properly select the cities and the satellite images for which the estimates are most accurate. Approaches to quantify $CO_2$ emissions from cities using satellite data have nonetheless emerged in recent years (Wu et al., 2018; Ye et al., 2020; Lei et al., 2021), giving interesting results. This is made possible by the launch of new 50   satellites (*e.g.* OCO-2/3, Sentinel 5-P, and GOSAT-2) measuring $XCO_2$ at kilometer resolutions with ppm accuracy.

     Schuh et al. (2021) use high-resolution simulations from an adaptive-mesh model, the OLAM model (Walko and Avissar, 2008a, b), to rank the largest cities of the world according to the ratio between the average amplitude of the anthropogenic signals over the city and the variability of the local background signal. This classification describes, a priori, for which cities the emission estimates will likely be the most accurate. This is achieved by using a single global model representing the influ-55   ence of large-scale variations in $CO_2$ concentrations. Danjou et al. (20xx) investigate a set of computationally-light methods





for estimating $CO_2$ emissions from a city using satellite images capturing most of the atmospheric $CO_2$ plume. This study extensively compares existing methods and their various parameterisation options at each step of the atmospheric plume detection and inversion process, using pseudo-images of $XCO_2$ concentration over Paris. This study leads to the identification of the the most suitable methods and configurations, among those tested, for the estimation of Paris $CO_2$ emissions. In parallel,

it quantifies the various sources of uncertainties associated with each method, at each step of the procedures. The errors in the emission estimates is most sensitive to the meteorological conditions, and more specifically to (i) the spatial variability in the wind direction and (ii) to the homogeneity of the background concentration field. However, this previous study considers only one city, Paris, thus applicable to a specific amplitude range in emissions, to a specific spatial distribution of the city emissions, to a single type of local topography, to a single type of background concentration field, and to mid-latitude meteorological con-

ditions. These results, both in terms of the distribution of the errors on the emission estimate (bias, IQR) and the sensitivities of this error to the spatial variability in the wind direction and in the background concentration field may, unsurprisingly, not apply to all cities.

Wang et al. (2018) evaluates the ability to estimate emissions from most urban areas ($\approx 5000$, whose contours are defined on objective criteria) and power plants with images similar to those of the CO2M mission. However, the emission uncertainties

estimates do not account for the errors in atmospheric transport. Their study only addresses the sampling (swath, cloud cover loss, spatial resolution) and accuracy limitations of the $XCO_2$ imagery. However, the uncertainty on the shape and position of the plume (and thus the meteorology and the characteristics of the cities) can also influence the results and thus the ability to estimate the emissions of a city.

We choose here to study a limited number of cities by applying the computationally-light methods of Danjou et al. (20xx) to

simulated $XCO_2$ fields by the atmospheric model OLAM. This experimental framework allows us to realistically account for the variations in background $CO_2$ concentrations and the uncertainty in the meteorology. This approach allows us to identify the main criteria of classification of the images based on the performances of the emission estimation, contrary to Schuh et al. (2021) which more simply approach the problem by the signal-to-noise ratio of the concentrations. Using the computationally-light methods best-suited for urban emission quantification, we will extend our analysis to various metropolitan areas, both

in terms of emission estimation performances and sensitivities. We make the assumption that the configurations chosen in the framework of Paris remain optimal for other cities. This assumption seems justified, as the chosen methods differ from the others on objective criteria, as described in Danjou et al. (20xx).

Such an analysis can help identify optimal targets for satellite targeting modes, for example, for OCO-3 SAMs, or, when processing large datasets from future imagers such as CO2M, help identify the portions of the data yielding images worth

processing for plume detection and inversion. In addition, our analysis can help to robustly assess the errors associated with urban emission estimates as a function of city type and atmospheric or observational conditions. At a minimum, we hope to indicate the reliability of the inversions, making it possible to validate or not the various estimates.

The section 2 briefly describes the configuration of the OLAM simulations used (section 2.2) and details the construction of the pseudo-images (section 2.3) and the emission zones we target (section 2.4). The inversion methods are described in

Danjou et al. (20xx) in their optimal configurations and are recalled in Section 3. Section 4 describes the decision trees used





to define the discrimination criteria. Section 5 shows the error analyses of the emission estimates according to the different variables studied and the criteria defined. The limitations are finally discussed in Section 6 along side an extension of the criteria distribution study to cities with populations over 1 million inhabitants.

## 2 Simulations of XCO$_2$ images over multiple cities

### 2.1 OLAM

The Ocean Land Atmospheric Model (OLAM) is a coupled ocean-atmosphere general circulation model (Walko and Avissar, 2008a, b) with a dynamical core that has been used in the Dynamical Core Model Intercomparison Project (DCMIP, Ullrich et al. (2017)). The main feature of the OLAM model is its hexagonal grid whose size is adaptive (see illustration on Fig. 1), which makes it possible to bring high resolution to the zones of interest (non-hydrostatic mesoscale) while maintaining a coarse

mesh over the rest of the globe (hydrostatic model). This approach reduces the need for computational time while maintaining a global domain. The transport modelled between the different regions uses physical and dynamical schemes that vary according to the resolution, in particular for submesh convection. Turbulent diffusion is parameterised using the Smagorinsky model, which depends directly on the resolution of the mesh. For the submesh convection, the cumulus clouds, the precipitation and the mixing are represented with a hybrid approach combining aspects of both Grell and Dévényi (2002) and Grell and Freitas

105 (2014).

The vertical levels are at constant altitude and can therefore cross the surface. The fact that the levels can cross the surface is optimal to avoid gradient errors on steep slopes that can be present in a pressure coordinate (or hybrid) grid (Ullrich et al., 2017). The adaptive horizontal grid allows areas with locally complex dynamics, such as mountainous or coastal areas, to be modelled at higher spatial resolution. It also allows to reduce the representation of urban plumes while limiting the computation

time compared to a global model with a single mesh size. Thus, over the selected cities, the mesh size is of the order of two to three kilometre and progressively enlarges until it reaches several degrees over the oceans.

### 2.2 Simulation of CO$_2$ transport

The OLAM atmospheric transport model is used to simulate the meteorological and CO$_2$ fields needed to build our pseudo-images. The fluxes from the CarbonTracker 2017 global inversion system (Peters et al., 2007) are used as model input for the

biogenic CO$_2$ surface fluxes. Anthropogenic emissions from the ODIAC spatialized inventory (Oda et al., 2018), are used to represent cities, industries, and powerplants. No temporal profile is applied to the ODIAC data, which means that the simulated anthropogenic emissions are constant over the month. From these data, the model will simulate on its hexahedral grid the wind, pressure, relative humidity and temperature fields (necessary for the calculation of the PBL height, via the calculation of the potential temperature field and the Nielsen-Gammon et al. (2008) formula) and the CO$_2$ concentration fields in the

atmosphere. The results of these simulations are then projected onto a regular grid to simplify the analysis of the model outputs. The simulations were done for 31 cities. We retrieve the 2D fields of XCO$_2$ (i.e. the integrated profiles of CO$_2$ weighted by





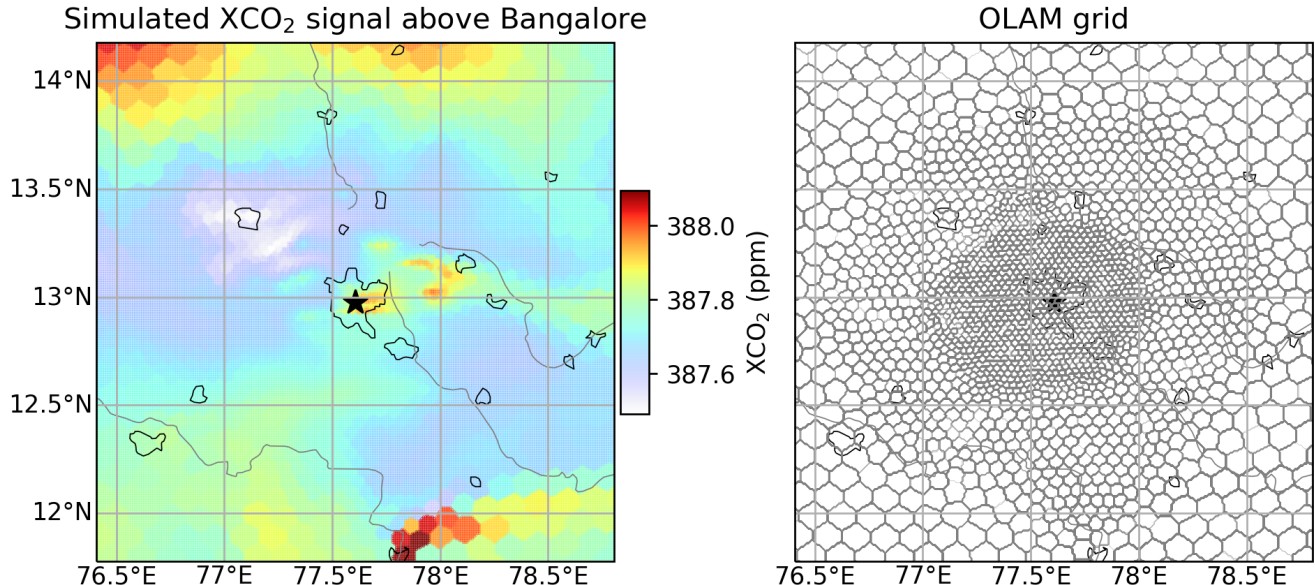

**Figure 1.** Illustration of the simulated $XCO_2$ signal above Bangalore on August 8th 2015 at 11a.m. (left panel) and the horizontal grid used for the simulation (right panel). The size of the illustration ($\approx$300×300km) is twice the size of the pseudo-images used in the study ($\approx$150×150km).

pressure levels) as well as the 3D fields of pressure, wind, relative humidity and temperature on the regular grid for all cities and their surroundings.

### 2.3 Pseudo-image generation

The original $XCO_2$ fields on a variable resolution grid were interpolated to a regular grid at approximately 1km×1km resolution, which is comparable to that of the OCO-3 SAMs (1.25km×2.5km, Eldering et al. (2019)) and that planned for CO2M (2km×3km, Sierk et al. (2021)). This resolution is slightly finer than the lowest resolution of the model's adaptive native hex grid, whose smallest mesh is a hexagon with 3km sides. Away from the city, the pixels are larger (hexagon up to 25km on a side). Therefore, the simulated patterns have a spatial resolution which is slighlty coarser than the spatial grid on which the

analysis will be conducted. The simulations cover a little more than one month (08/01/2015-09/10/2015), providing hourly $XCO_2$ fields. For each day of the simulated period, we retain the hourly fields of $XCO_2$ between 10:00 and 17:00 local time for our pseudo-images. This simulated database corresponds to a total of 9920 images interpolated at 1km×1km resolution. The spatial extension of the pseudo-images is restricted to a 150-km square whose axes follow the meridians and parallels and whose centre is the barycentre of the targeted city (in terms of $CO_2$ emissions). This size is close to that of Danjou et al. (20xx)

and is halfway between that of the OCO-3 images and that expected for CO2M. Finally, a random noise of 0.7 ppm standard deviation is added to the simulated $XCO_2$ field to simulate the satellite pseudo-data. This value corresponds to the target accu-





racy for a single $XCO_2$ measurement from the $CO_2M$ mission, similar to the current precision of $XCO_2$ measurements from OCO-2 (Worden et al., 2017).

## 2.4 Defining the boundaries of the cities

The first task for urban plume inversions is to define the targeted emission zone. Our definition of the emission zones is based on approximate considerations regarding the size of plumes that can be detected in a SAM and on an identification of the most emitting pixels from the spatialized inventories (using a similar concept but a different and more straightforward approach compared to Wang et al. (2019)).

Since the typical size of a SAM is 80km×80km, we set the size of the targeted emission zone at roughly the size of a 20km
radius circle. Thus, the emission zone we target occupies around 20% of a typical SAM. To define the boundaries of a emission zone, we first set its centre at the barycentre of anthropogenic emissions within the pseudo-image. We then restrict ourselves to a disc of 50km radius around this centre. The size is arbitrarily fixed at 2.5 times the 20km-long targeted emission zone radius. Within this 50km-radius disc, we select only a fraction ($1/2.5^2$) of the most emitting pixels of the $XCO_2$ pseudo-images. This fraction is explained by our choice to work with target areas of about $\pi \times 20^2$ km$^2$, i.e. $1/2.5^2$ of the surface of the 50km radius
disc in which the analysis is performed. In order to form a spatially coherent set, we extend the selected area to all pixels within 5km of one of the pixels retained by this first selection. This enlargement allows us to avoid complex cuttings of the city and to obtain groups of pixels where emissions are statistically high. The last two steps include (i) the selection of the sole cluster of pixels located above the city centre and (ii) the addition of pixels not categorised as belonging to the target area but completely surrounded by the target area. The final target area covers an area between 1333km$^2$ (Lahore) and 2063km (New-York) which
corresponds to 20-33% of the spatial coverage of most OCO-3 SAM images ($\approx$80×80km). We will call this targeted emission zone "the city" hereafter. Note that, the area with significant emissions may extend well beyond the administrative limits of the city, which justifies our choice.

It should be noted that the OLAM simulations used here separately track atmospheric signals from metropolitan areas (i.e. the metropolitan area "plume") for each of the 31 cities concerned and the atmospheric signal from anthropogenic emissions
outside these metropolitan areas. In theory, this separate monitoring, which is very costly in terms of computing time and resources, allows a detailed analysis of the detection and inversion capabilities of the plume, such as that carried out in (Danjou et al., 20xx). The boundaries of the metropolitan areas are defined in the OLAM simulations using the GRUMP v1.0 database (for International Earth Science Information Network CIESIN Columbia University et al., 2011). Unfortunately, these boundaries exceed the typical size of a SAM in most of our cases, which prevents us from using these boundaries to define the
emission zones and, therefore, from exploiting the separate simulation of the metropolitan area plume in our analysis.

## 3 Inversion methods

One of the methods tested by Danjou et al. (20xx) is based on the comparison of the urban plume detected in the image to a straight Gaussian plume. This comparison requires many preliminary steps. First of all (i) the definition of the boundaries





of the urban area whose emissions we want to estimate. The method used here is described in section 2.4. Second, (ii) the plume boundaries are defined by the pixels located above the city and those in the cone downwind of the city within an angle of 45°. The wind used to define the orientation of the cone is the average wind direction in the PBL over the entire image (from the OLAM simulation). Once the boundaries of the plume are known, we (iii) estimate the background concentrations, i.e. the $XCO_2$ signal in the plume which is not generated by the city emissions. This background concentration is extrapolated from the $XCO_2$ values of pixels outside the plume using a Gaussian kernel. The difference between the $XCO_2$ pseudo-image and the estimated $XCO_2$ background leads to an estimate of the plume generated by the city emission. We then (iv) calculate the central axis of the plume using a degree 5 polynomial regression using the pixels in the plume, weighted by the estimated $XCO_2$ signal from the city, as described in Danjou et al. (20xx). Using this central axis of the plume, we (v) delineate the area of the plume that will be used for the Gaussian plume optimization (*analysis area*). This area is located between one times the approximate radius of the city ($\approx$20km) and one and a half times the approximate radius of the city ($\approx$30km) along the central axis of the plume (the justification for these distances is given later in the paragraph). At this stage, we have extracted the estimated $XCO_2$ signal from the city and we have determined the pixels that we will use for the optimization. We estimate the effective wind (vi), i.e. the wind driving the $XCO_2$ plume from the city, using the averaged wind within the PBL and within the analysis area. Finally, we estimate the emissions (vii) by inverting the following formula as defined by Krings et al. (2011); Danjou et al. (20xx):

$$\Delta\Omega_{gp}(x,y) = \frac{F}{\sqrt{2\pi} * |\boldsymbol{W}| * \sigma_y(x)} e^{-\frac{y^2}{2*\sigma_y(x)^2}} \tag{1}$$

where the x and y axes follow the directions parallel and perpendicular to the effective wind, $\boldsymbol{W}$ is the effective wind vector, F is the whole-city emissions estimate, and $\Delta\Omega_{gp}$ is the mass of $CO_2$ in the atmospheric column per unit area. The term $\sigma_y(s)$ accounts for the horizontal extension of the source. We take $\sigma_y(x) = a * (x + (\frac{r}{4a})^{1/0.894})^{0.894}$ as Krings et al. (2011), where $a$ is the Pasquill stability parameter (Pasquill, 1961) and $r$ the city radius.

To estimate the emission budget, we perform a minimization of the mean square differences between the modelled mass per unit area ($\Delta\Omega_{gp}$) and the observed mass per unit area. The observed mass per unit is calculated from the $XCO_2$ signal from the city derived in step (iii) using : $\Delta\Omega(x,y) = \frac{M_{CO_2}}{M_{dry\ air}} * \Delta XCO_2 * 10^{-6} * \frac{P_{dry\ air}(surface)}{g}$, where $g$ is the Earth's gravity and $P_{dry\ air}(surface)$ is the dry air surface pressure.

The emission budget $F$, the Pasquill parameter $a$, the city radius $r$ and the orientation of the axis (i.e. the wind angle) are optimized during this minimization. The initial values are: for $a$, the value given by the Pasquill (1961) table corresponding to the meteorological conditions at the time of the image acquisition; for the orientation of the reference frame, the direction of the average wind in the PBL (noted $\theta^{init}$); for the radius of the city, the average radius of the city (noted $r^{init}$). The choice of the initial value of the emission budget (noted $F^{init}$) is more critical. Indeed, setting an initial value close to the exact value (let alone the exact value) might artificially improve our results. Instead, we take a random number from a beta distribution (with $\alpha$=1.35, $\beta$=2.5 and a scaling factor of 5) multiplied by the actual emission of the central urban area, following Danjou et al. (20xx). We normalise the variables for the optimisation as follows: $X = \begin{pmatrix} \frac{F-F^{init}}{F^{init}} & \frac{a}{120} & \frac{\theta-\theta^{init})}{\pi/4} & \frac{r-r^{init}}{r^{init}/2} \end{pmatrix}^T$. We





further impose bounds on these variables during optimization (the bounds are shown without the normalization for clarity) $F \in [-F^{init}; +\infty]$, $a \in [0; 240]$, $\theta^{init}[\theta^{init} - \pi/4; \theta^{init} + \pi/4]$ and $r \in [0.5 * r^{init}; 1.5 * r^{init}]$.

The methods used for steps (ii) to (vii) are those defined as optimal by Danjou et al. (20xx). Step (i) have been redefined here, ans steps (ii) and (v) have been slightly adapted. We choose to make the analysis area (step v) closer (and smaller) than in Danjou et al. (20xx). The new analysis area is located between the edge of the city ($\approx$20km) and 1.5 times the radius of the city ($\approx$30km) along the plume centerline, while it was located between the edge of the city ($\approx$20km) and 2 times the radius of the city ($\approx$40km) along the plume centerline in Danjou et al. (20xx). The conclusions on the sensitivity of the inversions to the analysis area indicate in Danjou et al. (20xx) that the closer the analysis area is to the city, the better the estimate. The plume definition method (step ii) has been slightly adapted compared to Danjou et al. (20xx). The emission area targeted for Paris in 4.3 (core urban area) was almost a disc of radius 20km, which allowed to define the plume as the 20km disc centred on the centre of Paris and the area downwind of this disc, using the mean wind in the PBL. However, due to the specific definition of the targeted emission zone here, most of the cities defined by step (i) have shapes that cannot be approximately described as a disc (see Fig. A1 in annex A), which requires an adaptation of the plume definition in step (ii). The plume is therefore denoted as the pixels directly above the city (and no longer in a disc of radius the approximate radius of the city) and the pixels downwind of the pixels directly above the city, according to the mean wind in the PBL.

Three additional emission estimation methods are retained by Danjou et al. (20xx): one using a rotating Gaussian plume model, another based on a mass balance in the plume (IME method) and the last based on a calculation of fluxes in cross sections of the plume. All three methods were tested, together with the one presented above, in this study. However, as the results are very similar for all 4 methods, our presentation of the results and analysis focus on a single method, the classical Gaussian plume, for clarity. The results from the other methods are presented in the appendix.

## 4 General principle of the analysis of sensitivities to observation conditions

We test here two types of variables: (i) predictable variables, used to determine the most favorable conditions for the inversion, which aggregate information about weather conditions and city characteristics; and (ii) diagnostic variables, used to evaluate the inversion results, which aggregate image diagnostics and inversion diagnostics. The sensitivity of the emission estimation error to the predictable variables in the first instance, and to the diagnostic variables in the second instance, are analysed in the same way. The analysis described below is therefore applied to each of the two groups.

As a starting point, we examine separately the relationship between each variable of the chosen group (predictable or diagnostic) and the error on the emission estimate. This preliminary analysis provides a first overview of the variables to which the error is sensitive. After this first step, we analyse all the relationships between the variables and the error to identify the one or two variables to which the error is most sensitive. This identification is performed using a decision tree, the depth of which determines the number of variables identified. The decision tree directly defines thresholds on these variables: following a strict interpretation of the algorithm, these thresholds can be used in a binary way to define whether a pseudo-image is suitable for emission estimation or not. In a more general way, these thresholds can be used as an indicative criteria to evaluate





the pseudo-images and the corresponding urban emission estimation. These identified variables, together with their respective thresholds, can be used to indicate the level of error of an estimate obtained during an inversion.

## 4.1 Preliminary analysis

To quantify the sensitivity of the error on the emission estimate to a specific meteorological variable, or a variable diagnosed by image processing or by inversion, we order our pseudo-images according to the values of the variable. For the analysis with predictable (respectively diagnostic) variables, we separate our set of 9,920 pseudo-images (resp. 4,259, cf. section 5.2.1) thus ordered into subsets of 496 (resp. 213) pseudo-images (5% of the total number). For example, when considering the mean wind in the PBL (i.e. a predictable variable), the first subset will include 496 pseudo-images corresponding to the 496 smallest values of the mean wind in the PBL. The second subset will be composed of the 496 images corresponding to the values of the mean wind in the PBL ranked between the 497th and the 992nd position. The last group of images will include 496 pseudo-images corresponding to the 496 largest values of the mean wind in the PBL. We then plot the error distribution for these subsets as a function of their rank to see if a significant trend is observed (see Fig. 3 for an example).

The simulations we use are based on an inventory of anthropogenic emissions with no temporal variations. As a result, the variables related to the emissions and the shape or topographical environment of the city have no temporal variability and therefore take only 31 values. Our study of the sensitivity of the error to these variables is therefore based directly on the analysis of the error distribution as a function of the value taken by the variable of interest.

## 4.2 Analysis with the decision tree learning algorithm

In this study, we seek to objectively determine simple criteria on which to base our decision as to whether or not we can be confident in the result of an inversion. A decision tree algorithm is well-suited to separate a population into several homogeneous groups according to a set of discriminating variables. In our case, we wish to have homogeneous groups with respect to the error in the emission estimate. To define these groups, we will use thresholds on the values of the predictable/diagnostic variables (see Fig. 2 for illustration and definitions).

### 4.2.1 Description of the decision tree learning algorithm

The tree is constructed following a recursive process: in the first step, the algorithm will separate a set of pseudo-images (at the *root node*) into two subsets (which will be the *child nodes* of the *root node*) using a threshold value for one of the variables of interest, so as to minimise the variance reduction; the algorithm will then separate the 2 subsets (the 2 *child nodes* of the *root node*) according to the same process, and so on. The recursion ends when the maximum depth set by the user is reached or when all the subsets produce the same value of the target variable, or when the separation no longer improves the result of the variance reduction. The algorithm is a brute-force algorithm: to determine the variable of interest used for the separation and its threshold, it tests all values of all variables. The variables of interest are, in our case, the predictable variables in a first analysis, and the diagnostic variables in a second analysis. The target variable is the absolute value of the relative error on the







**Figure 2.** Illustration of how a decision tree works. The decisions (orange ellipses) and the conditions (pink rectangles) are called *tree nodes*. The first condition, through which the tree is entered, is called the *root node*; the terminal nodes (the decisions, represented by orange ellipses) are the *tree leaves*. Nodes that are not leaves are called *internal nodes*. The path followed by an individual (i.e. the conditions that have been tested for that individual) to arrive at a leaf is called a *decision path*. The length of a decision path is equal to the number of conditions tested on that path (i.e. number of internal nodes traversed). The *tree depth* is the length of the longest decision path. The input pseudo-image will follow the decision path in bold and will classified in the bold leaf.





emission estimate. The maximum depth of the tree is set to 2 and we impose that the leaves must contain at least 10% of the starting set. The starting set (at the *root node*) is described in the following paragraph.

### 4.2.2 Description of our method for determining the decision criteria

A simple approach is to use the total set of pseudo-images (9920 pseudo-images in the case of the analysis of predictable variables, ≈4259 pseudo-images in the case of diagnostic variables) as the input set for the decision tree learning method. We thus obtain at most 4 subsets and select the one with the smallest Mean Absolute Error (MAE) on the emission estimate. This subset is considered as the subset of pseudo-images best suited for emission estimation and the rest of the pseudo-images as less suited ones. We then study the distribution of the error for this set as well as the pair of criteria that led to this partition. In doing so, we have no information on the stability of the criteria and thresholds with respect to the starting set. This is problematic, especially since the city features can only take 31 values for the 9920 images, which increases the risk of overfitting.

To overcome this problem and to get an idea of the stability of the criteria, we create 100 sets of pseudo-images each composed of random samples of 10% of our total set of pseudo-images. We apply the learning algorithm to each of the 100 sets. We look at the subsets corresponding to each leaf and select the one with the smallest MAE. The decision path that leads to this leaf gives us the pair of criteria that we retain. This gives us 100 pairs of criteria. We analyse the redundancy of the criteria across these 100 pairs and the stability of the threshold values of the pair with the highest occurrence. The different threshold values found for the pair with the highest occurrence are applied to determine, for each pair, a subset of pseudo-images for which the emission estimates are accurate. The distribution of the criterions obtained with the 100 sets of images, as well as the error distributions of these subsets of pseudo-images are studied to determine the reliability of the criterion threshold values.

### 4.3 List of variables

We tested 15 predictable variables (8 characterizing the weather and 7 characterizing the city) and 10 diagnostic variables (1 being an image diagnostic and 8 being inversion diagnostics). The detailed list is provided in the appendix B. We will provide some examples here.

To characterize the meteorological conditions, we have, for example, retained the average wind speed in the PBL and the spatial variability of the wind direction (calculated as the circular variance of the 3D wind field in the PBL at the observation time), 2 variables whose influence on the accuracy of the emission estimation has been highlighted in the study of Danjou et al. (20xx). We have also looked at commonly-used quantities characterizing the wind (divergence, vorticity,.. of the wind in the PBL). To characterize the city properties, we looked at spatial variables (its size, the topographic variability in the surroundings, its symmetry) and variables representing the characteristics of the urban emissions (emission budget given by the inventory, emission density). In our pseudo-data experiments, the analysis are based on values of the predictable variables that are extracted from the model, i.e. on the "true" values for all predictable variables. When using real satellite images (which is out of scope of this study), the analysis will rely on estimates bearing uncertainties, which could decrease the potential to identify suitable observation conditions. We note here that during our evaluations, the thresholds given in section 5.2 will be compared to crude estimates when dealing with actual satellite data, a possible source of errors in the classification.



To characterise the background $XCO_2$ field when analyzing the image, we use the spatial variability of the $XCO_2$ concentration. This variable has been highlighted by Danjou et al. (20xx) as influencing the estimation of emissions. This is the only variable diagnosed directly in the image among the list of diagnostic variables investigated here. With real data, the size of the image and its spatial coverage may have an influence on the accuracy of the emissions estimate. In this case, including this size in the list of diagnostic variables would make sense. However, this is not the case as we are working with synthetic data and all our images have the same size. Reproducing this variability in the coverage of real data is outside the scope of this study. The diagnostics of the inversion robustness include the size of the plume, the residual error after the optimisation with the Gaussian plume, the curvature of the central axis of the plume, the ratio between the estimated amplitude of the city signal and the variability of the signal outside the plume,... Unlike predictable variables, the calculated values for the diagnosed variables are directly inferred from the observations with real data. Therefore we will not have classification errors due to this. However, the values taken by the variables might have incorrect distributions in this theoretical study. For example, the distribution we use to simulate the measurement noise in our simulations is much simpler than actual measurement errors (see Discussion of Danjou et al. (20xx)).

## 5 Results

### 5.1 Preliminary analysis

When we apply the GP2 inversion method to our 9920 pseudo-images, we obtain an emission estimate in 92% of the cases (i.e. for 9119 pseudo-images). The bias (median) of the error distribution on the emissions estimate is -16% of the emissions, and the spread of this distribution (IQR) is 78% of the emissions. Danjou et al. (20xx), in their synthetic data study on the city of Paris, defined an image discrimination criterion based on the spatial variability of the wind direction, with a threshold of 7°. When we apply this filter, we reject 46% of the 9119 pseudo-images and obtain a much less biased distribution of the error (5% of the emissions) and slightly less spread out (64% of the emissions). However, despite the application of this criterion, the variability of the error distribution remains large across cities. After filtering, the error distribution for the city of Lahore (largest MAE on the emissions estimate) shows a bias of -21% and a spread of 154% of the emissions, while that for Moscow (smallest MAE on the emissions estimate) shows a bias of -3% and a spread of 26%. This confirms that, although the criterion defined in Danjou et al. (20xx) is relevant, our filtering step does not seem to be sufficient to select the pseudo-images. The strong disparity between cities suggests that the city characteristics (topography or city-specific atmospheric conditions) and its emissions (spatial distribution, magnitude,..) also play a role in the error of the emission estimate.

As in Danjou et al. (20xx), emissions are strongly underestimated when the wind is weak or when the spatial variability of the wind direction is strong (see Fig. 3). These 2 variables are also strongly correlated here (spearman correlation of -0.75). The results are more accurate (lower bias and IQR) when the meteorological conditions favour the ventilation of the emitted $CO_2$ in a narrow and straight plume, i.e. with a high wind speed and a low variability of the wind direction; but when the emitted $CO_2$ accumulates above and in the vicinity of the city in a diffuse plume with high values of $XCO_2$, or forms a plume with a complex structure, the results of the emission estimation are impacted by significant errors.





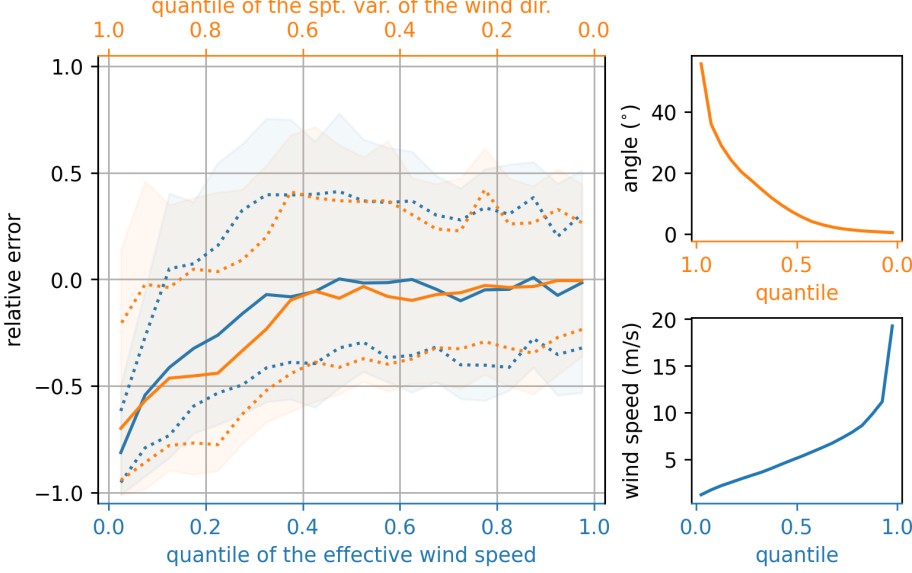

**Figure 3.** Sensitivity of the emission estimation error to two discriminant variables: the estimated effective wind speed (blue) and the spatial variability of the wind direction (orange). The left panel shows the evolution of the error distribution as a function of the quantile of the variable of interest: the solid line indicates the median, the dotted lines the 1st and 3rd quartiles, the highlighted area the quantiles at 15.9% and 84.1%. The left-hand panels show the values taken by the variables of interest for the different quantiles.

The error in the emission estimation also shows sensitivities to other variables characterizing the observation conditions: sensitivities to the emission budget, to the ratio between the average anthropogenic signal and the variability of the background signal, or to the difference between the optimised inversion angle and the average wind direction in the PBL are also visible
(see Annex C).

The error on the emission estimate thus shows sensitivities to several variables, some of which are correlated. These sensitivities can be complex, impairing our ability to determine the optimal set of thresholds and variables on the basis of sensitivity tests to define the most optimal pseudo-image discrimination criteria. However, the supervised learning method described in section 4.2 will allow us to determine the discrimination criteria in a more objective way, despite covariances among variables.

**5.2   Application of the decision tree method..**

**5.2.1   ... for predictable variables**

This first analysis, using the decision tree learning method described in section 4.2.1, is based on the results of the inversion of the 9119 pseudo-images produced by the GP2 method. We focus on the discrimination criteria given by our learning method with 100 different samples, as described in section 4.2.2. For 82 of the 100 samples, the pair of criteria given by our learning
method is the spatial variability of the wind direction and the emission budget, i.e. favoring large emissions and low variability





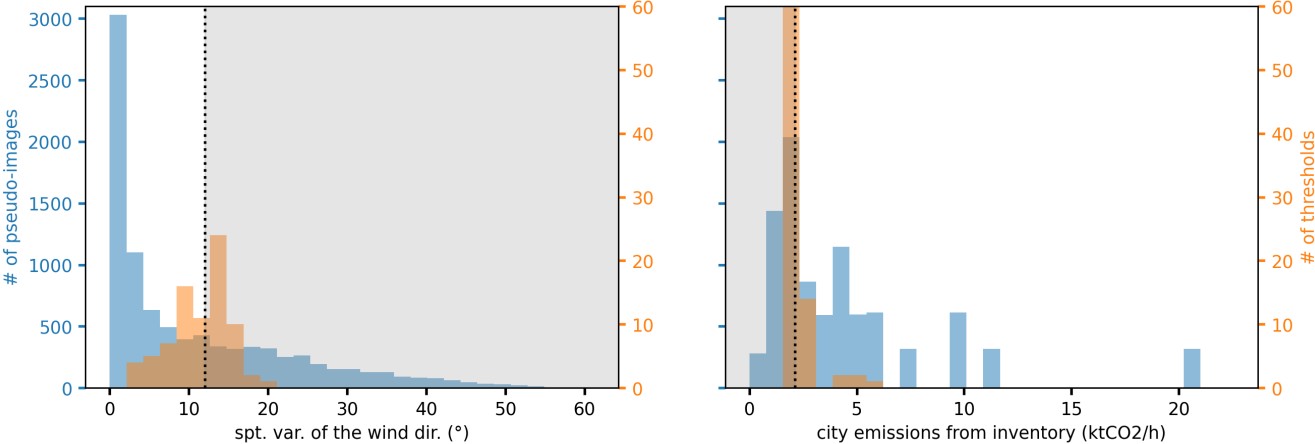

**Figure 4.** Distributions of spatial variability of wind direction (left) and city emissions (right). In orange the distribution is that of the criterion values and in blue that of the simulations. The black line indicates the median threshold found and the grey area the the discarded subset.

of the wind direction. For the remaining 18 samples, wind direction variability appears 9 times in the criterion pair and emission budget 7 times. The other variables appearing in the criteria pairs for the 18 samples are spatial variability of emissions in the city (5 occurrences), spatial variability of the emissions in the city (4 occurrences), mean PBL height (2 occurrences) and length of minor axis of ellipse (2 occurrences). For 6 samples, the pair of criteria is in fact a singleton indicating that one variable is
significantly more important than all the remaining variables. The spatial variability of the wind direction and the emissions within the city thus stand out very strongly.

We will now study in detail the threshold values taken by the spatial variability of the wind direction and the city's emission budget for these 82 pairs of criteria. The distribution of the threshold on the spatial variability of the wind direction is characterised by a median of 12° and an IQR of 5°. 10% of the inversions are found between the bounds formed by the quartiles
of this distribution (9° and 14°). The emission budget distribution is characterised by a median of 2.1ktCO$_2$/h=5.1MtC/yr and an IQR of 0.7ktCO$_2$/h. 22% of the situations fall between the bounds formed by the quartiles of this distribution (2.6ktCO$_2$/h and 1.9ktCO$_2$/h). The distribution of the criteria is therefore spread out, as illustrated by Fig. 4. The 82 subsets are, however, homogeneous in terms of the median of the error distribution (-7% [-6%,-8%]) and the IQR (55% [52%,58%]). This is less the case for the subsets size (45% [36%,52%] of the 9119 pseudo-images). For comparison, other studies such as Wang et al.
(2019) or Lespinas et al. (2020) have found lower thresholds (respectively 2MtC/year and 0.5MtC/year) leading to more precise estimates (uncertainties of less than 20%). But these studies, which both follow the same formalism, include fewer sources of error, which explains our higher threshold and uncertainties.

For the following analysis, we take the medians of the threshold distributions of our 82 retained pairs as the thresholds for these two criteria. The subset formed by the pseudo-images respecting these two criteria is characterised by a median error
of the estimated emissions of -7% of the city's emissions, an IQR of 56% and includes 47% of the 9119 pseudo-images. The subset formed by the pseudo-images that do not respect these two criteria is characterised by a median error of the estimated





emissions of -31% of the city's emissions, an IQR of 99% and includes 53% of the 9119 pseudo-images. The criteria therefore allow us to isolate the pseudo-images that are most suitable for inversion, as the pseudo-images that do not pass the criteria give highly biased estimates.

The discrimination criterion based on the spatial variability of the wind direction reduces the bias and the IQR of the error distribution, while the criterion based on the emission budget only reduces the IQR. Indeed, by applying only the discrimination criterion based on the spatial variability of the wind direction, we obtain for the subset passing the criterion a bias of the error of -5% and an IQR of 68% (-31% and 99% respectively for the pseudo-images not passing the criterion). Applying only the discrimination criterion based on emissions balance gives us, for the subset of pseudo-images passing the criterion, a bias of -16% and an IQR of 66% (-17% and 110% respectively for the pseudo-images not passing the criterion). Thus the criterion

based on the spatial variability of wind direction is a selection criterion (the pseudo-images that do not pass the criterion are considered unusable), and the criterion based on the emission budget is a discrimination criterion (the pseudo-images that do not pass the criterion will give a less accurate emission estimate).

### 5.2.2 ... for diagnostic variables

In this section, the set of pseudo-images used for the analysis is the set of pseudo-images (47% of our previous set) passing the criteria on the spatial variability of the wind direction variability and on the emission budget defined in section 5.2.1.

The pair with the highest occurrence (42 out of the 100 pairs) is the "ratio between the average anthropogenic signal and the variability of the background signal" and "spatial variability of the $XCO_2$ concentration outside the plume". For the other samples we obtain 20 different pairs. The spatial variability of the $XCO_2$ concentration outside the plume is also used in

the calculation of the estimated signal to background ratio. The two variables have a correlation of 0.34. We thus choose to reduce the tree depth to 1 and remove the ratio between the average anthropogenic signal and the variability of the background signal from our list of variables of interest. The choice of which variable to remove between the 2 is made on the number of occurrences across the pairs (54 for the ratio between the average anthropogenic signal and the variability of the background signal, 77 for the $XCO_2$ signal variability).

In this new configuration, 72 samples out of 100 give the variability of the $XCO_2$ signal as a criterion. The distribution of threshold values found for this criterion has a median equal to 0.72ppm and an IQR of 0.02ppm. 19% of the pseudo-images in the test set fall within the bounds formed by the quartiles of this distribution. By taking the median of this distribution as the discrimination criterion, we obtain two subsets which contain respectively 30% and 70% of the tested set and are characterised by biases on the estimation of emissions of -6% and -7%, and IQRs of 74% and 50%. This discrimination criterion reduces the

IQR but not the bias. However, the accuracy of this criterion is questionable: 50% of the values taken by the signal variability outside the plume are between 0.70 (which corresponds to the measurement noise) and 0.73ppm. A slight variation (0.01ppm) in this separation criterion has a strong impact on the error distributions of the two subsets. Moreover, the modelisation of the instrument noise (which have an important impact on the signal variability outside of the plume) is over simplistic in our work. We therefore choose not to retain this criterion.



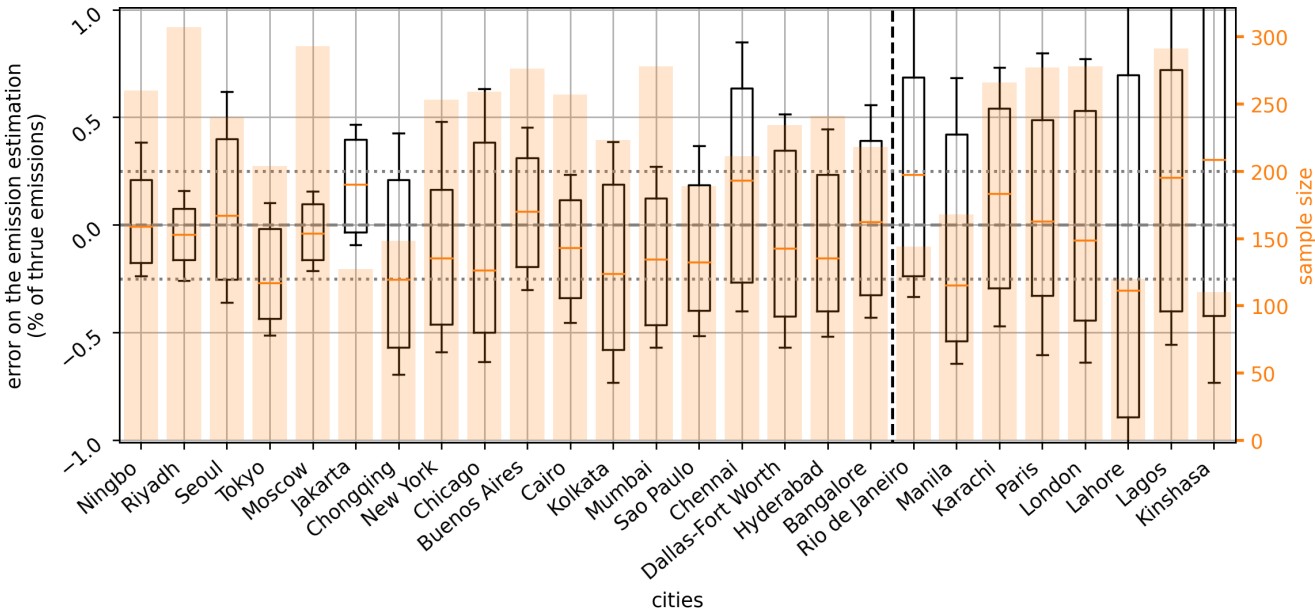

**Figure 5.** Distribution of the error on the emission estimates (boxplot) obtained with the GP2 method for the pseudo-images passing the selection criterion on the spatial variability of the wind direction ($> 12°$). The orange bars show the number of pseudo-images used. The dotted line separates the cities according to the discrimination criterion on the city's emissions budget with on the left the cities passing the criterion (emissions $<2.1\text{ktCO}_2/\text{h}$) and on the right the cities not passing it. The cities are ranked in descending order according to their emissions budget.

## 5.3   Study of the results by city

Of the 31 cities, five (Bogota, Lima, Los Angeles, Mexico City and Tehran) have more than 90% of the pseudo-images that do not pass the selection criterion based on the spatial variability of wind direction. We therefore have less than 30 images passing the selection criterion for these cities and choose to set them aside. All these cities are located in basins or at the foot of high mountain ranges, which explains the high spatial variability of wind direction for the vast majority of observations. Of the remaining 26 cities, 7 have their emission budget below the threshold of the emission budget criterion (see Fig. 5) and should therefore have low accuracy estimates. Paris is one of these cities in our simulations, with emissions of $1.8\text{ktCO}_2/\text{h}$ for the target area. The error distribution of the emission estimate for the city of Paris has a bias of 2% and an IQR of 83% for the pseudo-images passing the selection criterion on the spatial variability of the wind direction (86% of the pseudo-images). These results are close to those obtained in Danjou et al. (20xx) with the pseudo-images of Paris generated by WRF: the distribution of the error on the emission estimate had a bias of 4%, an IQR of 74% and 57% of the pseudo-images passed the criterion then defined. The IQR is larger in this study, and the number of images passing the criterion is higher. This can be explained by the





fact that the criterion was stricter in Danjou et al. (20xx) ($<7°$ in Danjou et al. (20xx), $<12°$ in this study), and that the selected months are not the same (December-April in Danjou et al. (20xx), August in this study).

Figure 5 shows that the accuracy of the estimate decreases with decreasing emissions budgets. However, this criterion alone
is not sufficient to classify the cities. In particular, the bias varies considerably from one city to another, even when their emissions are similar. Only 8 cities (Bangalore, Buenos Aires, London, Moscow, Ningbo, Paris, Riyadh and Seoul) have a distribution of error on their emissions estimates with a bias of less than 10%. Our selection allowed us to roughly filter out the worst situations for estimating emissions with our method, but not yet to fully understand the error dependencies. We want to point out that these errors are significant, even with many images ( 320) per city and our filtering. Future studies should
consider how best to use the emission estimates provided by satellite image analysis.

## 6 Discussion

### 6.1 Limitations of the study

Some potential sources of error not considered here (complexity of measurement error, loss of data due to - among other things - cloud cover and aerosols) have already been discussed in Danjou et al. (20xx) and are therefore not repeated here.
A major difference between the simulations in this study and those in Danjou et al. (20xx) is the lack of temporal variability of the emissions used. In reality, the plume is generated by the emissions that occurred up to a few hours before the satellite overpass, and inventories show significant daily cycles, in particular related to the traffic and industrial activity. When dealing with real data, our analysis zone may correspond to emissions that occurred, for example, three hours before the acquisition time, and comparing the emission estimate to the emissions at the acquisition time of the pseudo-image introduces an additional
error. The risk is therefore that we have optimistic error bars, as the temporal variability of emissions is not considered. In practice, Danjou et al. (20xx) showed that the analysis zones correspond to emissions that are very recent (less than 2 hours) in most cases. Thus, carrying out this study with variable emissions should not significantly alter our results, assuming that the emissions of a given city remain similar within two hours in the middle of the day (no morning and evening traffic rush hours). However, the issue of temporal variation of emissions arises with real data when we compare our emission estimates
with inventories. For cities without hourly emission budgets (or if the comparison is made with an inventory that does not vary on an hourly basis), we will have an additional source of error, this time coming from the estimated emission budget of the inventory.

The fact that one of the criteria is based on the city's emission budget may be problematic when using real data. Indeed, we will need an a priori value to rank the cities, which will be estimated from an inventory. Thus, relying on inventory emission
balances to rank cities could result in ranking errors.





## 6.2 Distribution of the discrimination criteria for the cities with more than 1M inhabitants

This section focuses on the number of cities passing the discrimination criteria for a non-negligible part of the year. We are interested in cities with more than one million inhabitants in 2018, according to UNDESA (2018).

**Figure 6.** Average annual cloud frequency over the period 2000-2014 derived from satellites Terra and Acqua (Wilson and Jetz, 2016) and location of cities with more than one million inhabitants (top panel). Distributions (4 panels from left to bottom) for cities of different continents of the pseudo-observation frequency with spatial variability of wind direction less than 11°, of the mean annual cloud frequency, of the emissions and of the proportion of sea surface in the neighborhood of the city (area within 30km of the city). The last panel (bottom right) shows the percentage of cities per continent passing the emission budget criterion, having less than 25% of their neighborhood covered by the sea, and for which the frequency of observations allowed a priori by the cloud cover and the threshold on the spatial variability of the wind direction is one day out of two (light orange), one day per week (orange), one day per month (dark orange).





The spatial variability of wind direction is calculated as the pressure-weighted circular variance of wind in the PBL in a
150km square centred on the city centre given by UNDESA (2018). For this analysis, the meteorological data (3D wind field,
pressure field, PBL height) come from the ECMWF ERA-5 product (Hersbach et al. (2018)) at 0.25° resolution for the year
2020. We calculate this variability for each day at 10 am, 1 pm and 4 pm local time. These different times are chosen to be
representative of the times when OCO-3 passes through. We calculate for each city the proportion of these "observations" for
which the spatial variability is above 11°. We can see in Fig. 6 that for the vast majority of cities (92%), this proportion is
above 50%. The distribution of the spatial variability of the wind direction is different from the one we have with the OLAM
model, where more cases were rejected. An explanation may be the much lower sampling of ERA-5 (around 25km against
around 3km for OLAM), which smoothes the wind direction variations. Nevertheless, we can see that, based on this variable,
the least suitable cities for emissions monitoring are located in Asia and America.

City emissions are evaluated with the ODIAC product for the year 2019 and using the definition of city boundaries defined
in Section 2.4. Among cities with more than one million inhabitants, only 40% pass this criterion.

Cloud cover is also a factor limiting the number of images that can be acquired, which is not considered in this study. Wilson
and Jetz (2016) is a database giving the frequency at which clouds cover a point on the globe. This dataset integrates 15 years
of twice-daily remote sensing-derived cloud observations at 1 km resolution. We are interested in the annual average of this
frequency to have an order of magnitude of the days not observable due to clouds. Figure 6 shows that the cloud frequency is,
for most cities, between 40 and 80%, whatever the continent.

Finally, the proportion of the water surface in the vicinity is also important for our measurements, as current instruments
cannot make measurements over both water and land in a limited time and space interval. We define the city neighbourhood
as the area within 30 km of the city edge as defined with the method described in Section 2.4. For most cities (77%), this
proportion is less than 25%.

To give an idea of the current ability to quantify urban $CO_2$ emissions using satellite imagery, we look at the distribution
of cities with emissions greater than 2.1ktCO$_2$/h and with less than 25% of the sea surface in their vicinity. We add an index
of how often we can measure them by multiplying the proportion of cloud-free days by the proportion of days where the
spatial variability of wind direction is greater than 11°. Figure 6 (right panel) shows the proportion of cities per continent
that pass the criteria and can be measured on average every other day, one day per week and one day per month (1 day/30).
Very few African cities (4 out of 57) pass our criteria, mainly due to their low emissions. The proportion of cities passing all
three criteria (emissions, sea in the vicinity, frequency of observation) does not change with the frequency threshold, as cloud
cover and spatial variability of wind direction are generally lower in Africa than in other continents. America and Europe show
similar results: most cities are rejected by our emission criterion, and the high cloud cover (often more than 50%) does not
allow for observations at least every other day. On the other hand, there are no more observable cities when the threshold on the
frequency of observation is raised from one day per week to one day per month. The observable cities in America and Europe
(30 cities out of 119 and 14 cities out of 58) can provide an order of magnitude of one observation per week if observed daily.
The seasonal distributions of cloud cover and spatial variability of wind direction are not taken into account in this analysis.
Asian cities, due to higher emissions, show a higher proportion of cities passing the criteria. Very few cities (16 out of 273) are





observable on average every other day. Again, the proportion of cities passing the criteria varies little between a threshold of
one day per week and one day per month (101 and 109 cities out of 273, respectively). Australia stands out: only five cities have
more than 1 million inhabitants. For this continent, the distribution of the variables we are interested in is fairly homogeneous,
which places them at the limit of observability with the criteria on emissions and the proportion of sea surface in the vicinity
(all the cities are coastal).

Asia and Australia stand out, with 36% and 40% of cities of more than one million inhabitants for which $CO_2$ emissions are
theoretically relatively easier to quantify than cities on other continents. They are followed by America and Europe, with 25%
and 24% of cities. Due to their lower emissions compared to other continents, African cities seem more difficult to monitor
(only 7% pass our criteria). These results are valid for imagers with characteristics close to those of the OCO-3 instrument
(2x2km resolution, 0.7ppm accuracy) and should be reviewed for future satellites with possibly better characteristics.

### 6.3 Other potential criteria

Wind speed is often cited as having an impact on the magnitude of error while quantifying greenhouse gases emissions of local
sources using satellite imagery (Varon et al., 2018, 2020; Nassar et al., 2022). As we have seen, using a criterion based on
wind speed is relevant, as low wind speeds are often associated with high spatial variability in wind direction. These situations
give rise to poorly ventilated plumes with complex structures whose corresponding emissions are difficult to calculate. This
study's decision tree learning method indicates that the criterion on spatial variability of wind direction is more accurate than
a criterion on wind speed with the set of images used here. However, this might be different when using real data. Indeed,
the resolution of the weather product used here is very high (3km, horizontally, xx vertical levels), higher than that of, for
example, the ERA-5 product ($\approx$25km, 37 vertical levels). With wind data at a resolution comparable to that of ERA-5, the
spatial variability of wind direction will be underestimated when the typical size of the horizontal variations is between 3 and
25 km, and the accuracy of the criterion will be lower. A criterion based on wind speed might then be more relevant, as this
variable is less sensitive to sampling.

Another criterion often associated a priori with error in emissions estimation is the ratio between the average anthropogenic
signal and the variability of the background signal (Schuh et al., 2021). This ratio quantifies the visibility of the plume and
indicates how easy it is to quantify the emissions. We have seen that the error on emission estimation shows a high sensitivity to
this variable (Section 5.1 and Annex C) and is apparent in our decision tree analysis for diagnostic variables (5.2.2). However,
this dependence of the error on the ratio "average anthropogenic signal" - "background variability" is slightly less important
in our analysis than the dependence on the background variability. The relevance of the background variability as a criterion
has already been discussed in section 5.2.2. A priori, we might have expected the error's dependence on this ratio to be greater
than its dependence on background variability. However, this dependence has already been partly filtered out by our analysis
of the predictable variables, with the criterion on the emission budget.



## 7    Conclusions

This study shows the performance of an automatic process for estimating urban emissions from XCO2 satellite images. This process is also independent of the target city: it is applied identically to all target cities. The methods used are low in computation time (on the order of a minute to process an image), which enabled us to process a database of around 10,000 images with a low failure rate (8% of the image). This study, therefore, contributes to the development of standard and automatic methods for operational applications to monitor urban emissions with satellite observation.

In this study, we have shown through a decision tree learning method that the spatial variability of the wind direction and the city's emission budget are the two main criteria, among those tested, to select the most suitable images for emission estimates at city scale and based on $XCO_2$ satellite imagery. We were also able to determine precise and objective thresholds on these criteria to select the images.

The threshold, of 12°, on the variability of the wind direction within the image area allows to reduce both the bias and the spread of the error distribution on the emission estimation. The threshold of $2.1 ktCO_2/h$ on the emission budget reduces the spread of the error on the emission estimate. The application of these two criteria simultaneously allows us to separate the pseudo-images into two sets: the first, grouping 47% of the pseudo-images, for which the distribution of the error on the estimation of emissions has a bias (median) of -7% of the emissions and a spread (IQR) of 56%; and the second for which the distribution of the error has a bias of -31% and a spread of 99% of the emissions. However, when we look at the results city by city, some of them show biases on emission estimates of over 10%, despite our filters. These significant remaining biases raise the question of the reliability of city-by-city studies as things stand.

This study has provided objective criteria for selecting the most suitable satellite images for urban plume inversion. However, we note here that these criteria are determined with synthetic data, estimated from atmospheric model simulations and inventories, previously evaluated against actual observations. With real data, these estimated values may differ from our current estimates, in particular for the emissions budget criterion. However, our study directly supports the interpretation of future inversion results using $XCO_2$ satellite images such as the OCO-3 SAMs.

*Code and data availability.*    Code and Data are available upon requests.

## A    Illustration of the boundaries of the target areas



**Figure A1.** Illustration of the OLAM boundaries of the cities and the boundaries of the target areas selected for the inversions. In blue are the boundaries of the areas we are targeting. The frame of the figures coincides with the area covered by the 150km square pseudo-images described in 2.3



# B Variables list





| analysis group | name | expression | type |
|---|---|---|---|
| predictable variables | mean wind in the PBL | $\frac{\sum_{\text{PBL}} \boldsymbol{W} * dP_{\text{air}}}{\sum_{\text{PBL}} dP_{\text{air}}}$ | meteorological conditions |
| | spatial variability of the wind speed | $\sigma(\|\boldsymbol{W}\|)_{\text{PBL}}$ | |
| | spatial variability of the wind direction | $\sigma(\theta_{\boldsymbol{W}})_{\text{PBL}}$ | |
| | PBL height | $< z_{\text{PBL}}^{\text{a.g.l}} >_{\text{pseudo-image}}$ | |
| | mean divergence of the 2D wind | $< \frac{\partial u^{2D}}{\partial x} + \frac{\partial v^{2D}}{\partial y} >_{\text{pseudo-image}}$ | |
| | mean vorticity of the 2D wind | $< \frac{\partial v^{2D}}{\partial x} - \frac{\partial u^{2D}}{\partial y} >_{\text{pseudo-image}}$ | |
| | mean stress of the 2D wind | $< \frac{\partial u^{2D}}{\partial x} - \frac{\partial v^{2D}}{\partial y} >_{\text{pseudo-image}}$ | |
| | mean shear of the 2D wind | $< \frac{\partial v^{2D}}{\partial x} + \frac{\partial u^{2D}}{\partial y} >_{\text{pseudo-image}}$ | |
| | *small axis* of the ellipse encompassing the city | ellipse defined with the library *opencv* | characteristics of the city (take only 31 distinct values) |
| | *big axis* of the ellipse encompassing the city | | |
| | variance of the city emissions | $\sigma(\text{emiss})_{\text{city}}$ | |
| | city area | $A_{\text{city}} = \sum_{\text{city}} 1$ | |
| | emission density | $\sum_{\text{city}} \text{emiss}/A_{\text{city}}$ | |
| | city topographic variability | $q_{90}\left(z_{\text{surf}}^{\text{a.s.l}}(x,y)\right)_{\sqrt{x^2+y^2}<50\text{km}}$ $- q_{10}\left(z_{\text{surf}}^{\text{a.s.l}}(x,y)\right)_{\sqrt{x^2+y^2}<50\text{km}}$ | |
| | emission budget | $\sum_{\text{city}} \text{emiss}$ | |
| diagnostic variables | spatial variability of the concentration in $XCO_2$ | $\sigma(XCO_2^{\text{obs}})_{\text{pseudo-image}}$ | image diagnostics |
| | estimated effective wind speed | $\|\boldsymbol{W}_{\text{eff}}\|$ | inversion diagnostics |
| | residual mismatch after optimisation | $\frac{\sqrt{<(\delta\Omega^{\text{model}} - \delta\Omega^{\text{obs}})^2>_{\text{panache}}}}{<\delta\Omega^{\text{obs}}>_{\text{panache}}}$ | |
| | optimized Pasquill parameter | / | |
| | optimized city radius | / | |
| | optimized plume direction | / | |
| | plume size | $\sum_{\text{panache}} 1$ | |
| | ratio "average anthropogenic signal" vs "background signal variability" | $\frac{<XCO_2^{\text{obs}} - XCO_2^{\text{bckg,calc}}>_{\text{a.z.}}}{\sigma(XCO_2^{\text{obs}})_{\overline{\text{panache}}}}$ | |
| | spatial variability of the $XCO_2$ outside of the plume | $\sigma(XCO_2^{\text{obs}})_{\overline{\text{panache}}}$ | |
| | curvature of the plume centerline | average distance between the centres of the cross sections and the linear plume centreline | |

**Table B1.** List of variables of interest divided by analysis group and type. The operators $< X >_E$, $\sigma(X)_E$, $q_{90}(X)_E$ and $q_{10}(X)_E$ denote respectively the mean, the standard deviation, the 9[th] decile and the 1[first] decile of the variable $X$ on the set $E$. The sets *a.z.*, *plume*, $\overline{plume}$, *city* and *pseudo-image* designate respectively the set of pixels in the analysis zone, in the plume, out of the plume, above the city and all the pixels of the pseudo-image. The reference frame used for the calculation of divergence, vorticity, stress, wind shear and topographic variability is the orthogonal frame with centre the centre of the city and with horizontal axis the mean wind direction in the PBL. $\boldsymbol{W}^{2D} = \left(u^{2D} \quad v^{2D}\right)^T$ is calculated as the vertical average of the wind in the PBL, weighted by the dry air mass.



## C  Extension of the study to other inversion methods

### C1  Inversion methods

Three other inversion methods have been optimized by Danjou et al. (20xx) and tested here: one based on the optimization of
a rotating Gaussian plume model (denoted GP3), one based on flux estimates across plume cross sections (denoted CS) and
one based on a $CO_2$ mass balance in the plume. Details of these methods can be found in Danjou et al. (20xx). Concerning the
pre-processing steps (i to vi, cf section 3), they are the same for the method based on a rotating plume model as those described
in section 3 for the straight plume model. For the other two inversion methods (CS and IME), the steps of defining the analysis
area and estimating the effective wind (steps v and vi) are different: the analysis area is the plume area within one time the
radius of the city along the central axis of the plume, and the effective wind is estimated with the wind tangent to the central
axis of the plume in the analysis area according to Danjou et al. (20xx).

### C2  General results

When we apply the CS, IME and GP3 inversion methods to our 9920 images, we get a result in over 98% of the cases. At
first sight, the error distribution on the emission estimate seems less biased for CS and IME (bias less than 10%) than for
GP2 and GP3 (bias between -13 and -16%). The IQRs of the error distributions are however larger for CS and IME (90-91%)
than for GP2 and GP3 (78-86%). When we apply the filtering of pseudo-images based on the spatial variability of the wind
direction denied in the Paris paper (i.e. pseudo-images for which the spatial variability of the wind direction is less than 7), the
underestimation of the emissions by the GP2 and GP3 methods disappears: the error distributions have a bias between -5 and
7% for the inversions based on GP2 and GP3, as well as for those based on CS and IME. The IQR of the distributions also
decreases: it is 75-76% for the inversions based on CS and IME and between 64 and 67% for those based on GP2 and GP3.
After filtering, we are left with 53% of the images. For CS, IME and GP3, like the results presented for GP2 in section 5.1, the
relative error on the emissions shows strong disparities between the cities, even after applying the Danjou et al. (20xx) filtering
based on the spatial variability of the wind direction.

Here we will detail the sensitivities of the error on the emissions estimate to the variables of interest. This qualitative study
is much reduced in the section 5.1 to keep the message concise and clear in the main body of the article.
As in the Paris study, the emissions are strongly underestimated when the wind is weak or when the spatial variability of
the wind direction is strong (see Fig. C1 (a) and (b)). The remark made in the Paris study is therefore still valid: the results
are more accurate (lower bias and IQR) when the meteorological conditions favour the ventilation of the emitted $CO_2$ in a not
very diffuse and straight plume, i.e. with high wind speed and low variability of the wind direction; but when the emitted $CO_2$
accumulates over and in the vicinity of the city in a diffuse plume with high $XCO_2$ values or forms a plume with a complex
structure, the results contain significant errors.





**Figure C1.** Distribution of the error on the emission estimate according in fonction of different variables of interest.

Figure C1 (c) shows a sensitivity of the error on the emission estimate to the actual emission budget. Despite noise, we can see that the IQR of the error decreases when the emissions increase. Cities with emissions have a plume that stands out more strongly from the background signal and allow a more accurate emission estimate. emissions.

The sensitivity of the error to the ratio of the average anthropogenic signal to the variability of the background signal is
shown in Fig. C1 (d) for the estimated anthropogenic signal and (e) for the actual anthropogenic signal. This actual signal-to-background ratio is close to that used by Schuh et al. (2021). The error sensitivities to these two ratios are very similar when this ratio is high, i.e. when the signal from the city differs most strongly from the variability of the background signal. In this case, the estimated anthropogenic signal is close to the real anthropogenic signal. The sensitivities of the error to these two ratios are however very different when these ratios are low. This can be explained by different reasons for the low ratios. For
the estimated background ratio, poorly defined plume boundaries lead to an overestimation of the background signal and thus





to a low estimated anthropogenic signal and an underestimation of emissions. For the actual background ratio, low emissions result in a weak anthropogenic signal that is difficult to discern, and thus a higher uncertainty in the emission estimate.

Finally, the error on the emission estimate is very sensitive to the radius of the city optimised during the inversion for the inversions based on a Gaussian plume (GP2 and GP3, see Fig. C1 (g)). However, when we apply the pseudo-image filtering based on the spatial variability of the wind direction defined by Danjou et al. (20xx), this sensitivity almost disappears. When the spatial variability of the wind direction is large, a dome, or at least a very diffuse plume, forms over the city and disturbs the optimisation of the city radius.

The error on the emission estimate thus shows sensitivities to several variables, some of which are correlated. These sensitivities can be complex and it is difficult at this stage to determine on the basis of these sensitivities which of the variables are the most sensitivities which of the variables are the most discriminating, and thus to determine the criteria for discriminating the most optimal pseudo-image discrimination criteria.

## C3   Application of the decision tree method

### C3.1   ... for predictable variables

The application of our learning tree method to inversions with GP3 gives very similar results to those described for GP2 similar to those described for GP2 in section 5.3.4.2 (see table 5.2). The pair of criteria that emerge is the same (spatial variability of wind direction and emission balance) with a slightly higher number of occurrences (95 for GP3, 82 for GP2). For the inversions with CS and IME, the same pair of criteria is also found, but with a lower number of occurrences (53 and 63 respectively).

The distributions of threshold values for the criteria are very similar for all methods. The medians of the thresholds found for the spatial variability of wind direction are $10°$ for the GP3 method, $10°$ for the CS method and $11°$ for the IME method. For the emissions bugdet, they are $1.9 ktCO_2/h$ for the GP3 method, $2.0 ktCO_2/h$ for the CS method and $1.9 ktCO_2/h$ for the IME method.

The bias ($<10\%$) and IQR (between 52% and 70%) on the emission estimate for the subsets of pseudo-images passing the criteria are close for the different inversion configurations (cf. table 5.2), as well as the size of these subsets (between 36% and 55%). The subsets that do not pass the discrimination criteria show differences depending on the inversion configuration. The results with GP3 are similar to those with GP2 in terms of bias, IQR of the error on the emission estimate and plume size; but for CS and IME the biases are smaller (between 6 and 12% in absolute value for CS and IME, between -25 and -37% for GP2 and GP3) and the IQRs are larger (higher than 120% for CS and IME, lower than 107% for GP2 and GP3).

### C3.2   ... for diagnostic variables

In this section, the set of pseudo-images used for the analysis is the one formed by the pseudo-images passing the criteria on wind direction variability and on the emission balance. The inversion methods (GP3, IME, CS) are tested separately. For all these methods, no pair of criteria has more than 40 occurrences when the tree depth is set to 2. We therefore also reduce the tree depth to 1. Plume size appears as the main criterion for IME and CS, with respectively 44 and 42 occurrences, without





standing out strongly (appears for less than half of the samples). For GP3, the error of the optimisation appears as the main criterion, without standing out here too (42 occurrences). We therefore choose not to retain these criteria.

**C4    Study of the results by city**

The thresholds for the selection criteria used are those found for the GP2 method (cf section 5.2.1). We have the same 5 cities (Bogota, Lima, Los Angeles, Mexico City and Tehran) for which more than 90% of the pseudo-images do not pass the selection criterion based on the spatial variability of the wind direction.

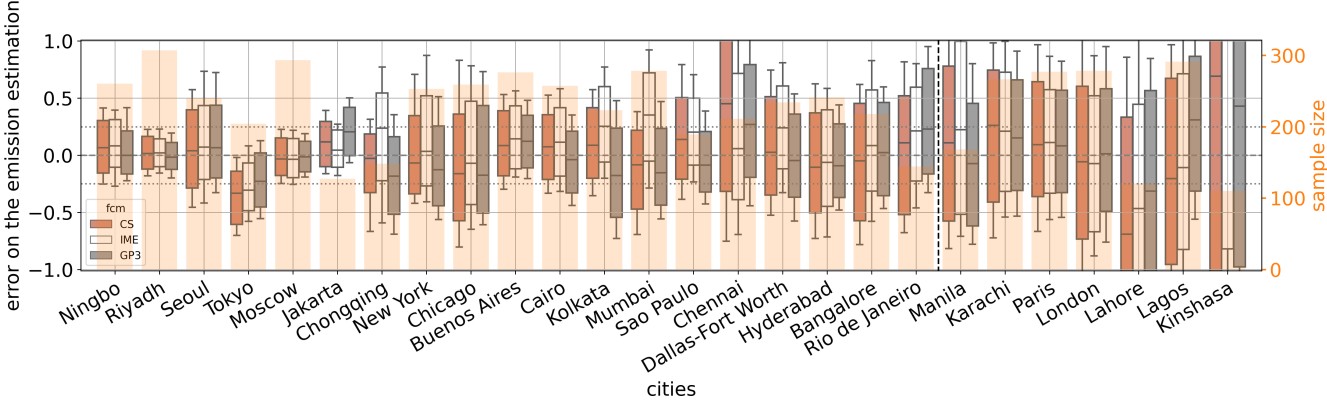

**Figure C2.** Distribution of the error on the emission estimates (boxplot) for the pseudo-images passing the selection criterion on the spatial variability of the wind direction ($> 12°$). The orange bars show the number of pseudo-images used. The dotted line separates the cities according to the discrimination criterion on the city's emissions budget with on the left the cities passing the criterion (emissions $< 2.1$ktCO$_2$/h) and on the right the cities not passing it. The cities are ranked in descending order according to their emissions budget.

As with the GP2 method, the accuracy increases globally with the cities' emissions (see Fig. C2). However, here again, this
parameter does not fully explain the disparity of the results between cities.

*Author contributions.*  AD performed the data analysis and wrote most of the manuscript. AS ran the OLAM simulations, provided helpful explanations on the outputs, wrote part of section 2.1 and 2.2 and was at the origin of section 4.3 with its pertinent comments. GB and TL closely supervised the redaction, took part in the design of the data analysis and improved the quality of both the scientific content and the level of the manuscript by their careful reviews. FMB supervised the work and gave useful remarks on the study and the manuscript.

*Competing interests.*  The authors declare no competing interests.



*Acknowledgements.* This work was supported by the CNES (Centre National d'Etudes Spatiales), in the frame of the TOSCA OCO-3-city project.





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
