# Peer review of "Optimal selection of satellite $XCO_2$ images over cities for urban $CO_2$ emission monitoring"

_Atmospheric Measurement Techniques, 2023_

## Referee Comment (RC1)

**Review of Danjou et al., AMTD:**

In this manuscript, the authors discuss a methodology to estimate $CO_2$ emissions by cities in an automatic way to validate self-reported emissions by cities. For this, they use $CO_2$ concentrations simulated by a global model with an adaptive horizontal resolution enabling them to increase the resolution to about 23 km$^2$ locally around 31 cities with improved representation of the meteorology and potentially better comparability to satellite measurements around these targets. Their methodology consists of calculating hourly $XCO_2$ during local daytime from their simulation in a 150x150 km$^2$ region around the city and then derive criteria to select images which they conclude will be appropriate for satellites to observe the emission plume using instruments like OCO-3 or CO2M. While it is an important scientific topic the authors try to tackle, the descriptions in the manuscript are confusing and have many imprecisions making the manuscript hard to follow, which is why I suggest major revisions before the manuscript should be even thought of being published. In addition, it seems that the manuscript has been submitted in a preliminary stage because almost no abbreviations have been defined, some values are "xx" (e.g., L496) and there are many typos in the figures. My comments are separated in general and specific comments, followed by technical corrections.

**General comments:**

1. Citation Danjou et al. (20xx): While it is appreciated that it is made clear by the year number 20xx that it is a preprint under review, the whole manuscript is based on this non-published study (cited 40 times). In order to make the results of this current manuscript understandable for the general public, it is important to add a comprehensive overview with **all** the results from your previous study needed to understand the results in this manuscript. In addition, please add the journal the manuscript has been submitted to so that at least it may be found in the future. You could also consider to publish the preprint in a citable space and cite this here, which would make this manuscript here much more transparent at this stage.

2. It is never defined which is your reference determining the "error" which is mentioned at many points in the manuscript (e.g., L7, L91, L362 and many others). Unless the reference data of the city emission is not clear, the whole error analysis does not make sense, so please describe clearly what you use as a reference at some point of the manuscript. See also specific comment for L362. In addition, this also means that the motivation of this study is not clear: Are you investigating how well suited different methods are to determine emissions by cities (model as "true" emissions and method as uncertainty) or are you interested in which cities and meteorological conditions are best for analyzing city emissions with satellites, which is suggested by the title but where it is unclear what is your reference?

3. A major issue with satellite observations are clouds which will decrease the coverage. They are not mentioned in the methodology description (Sect. 4) in any way, but will probably be the main limitation to the emission estimates. I'm sure you can derive cloud information from your simulations which in my opinion has to be the first criterion to select whether emissions can be estimated using satellite measurements. This cloud screening would be something like a pre-selection of the images. Otherwise, the selection of the images is not "optimal" as suggested by the title.

4. It seems as if it is not accounted for detection limits of the satellites at any point in the manuscript and should be considered or at least mentioned somewhere. Can a satellite like OCO-3 or CO2M even distinguish emissions in the order of 2.1kt/h from the background?

5. All the results are based on simulations by the OLAM model, but the description of the model and of the performed simulations are not comprehensive, many things are missing making it hard to understand, see specific comments below for lines 95 to 138 and for Fig. 1.

6. You are mentioning at various points in the manuscript that you are interpolating and extrapolating without including information which methods are used and why you are using them. Do these interpolations to a 1x1 km grid influence the simulated emission fluxes? Are they mass-conserving? Why and where are you extrapolating the $XCO_2$ values (L173)? Common practice is to use the average of the background region. What is the reason that you use another procedure?

7. I am missing a description how you suggest to use your method for real measurements. How do you suggest to derive the variables needed for the analysis with your method with respect to real satellite measurements?

8. I would prefer reordering the methodology section because it is hard to follow in the current setup. First, I suggest putting the figure and table from Appendices A and B (Fig. A1 and Table B1) to their place in the main text where they are discussed as they have not any description in the appendices anyway. Second, it would be much better to move the descriptions from Sect. 5.2 to the decision criteria selection part in Sect. 4.2 and move the list of variables before this. This would improve the readability because otherwise, the reader is left with the methodology without the outcome in the current Sect. 4.2.

9. At some points you basically say that this is still work in progress and that your method is not applicable to all cities (L320, L324, L527) because it may depend on the surroundings of each individual city, which is okay but then the abstract should provide this information.

**Specific comments:**

- L1: define XCO2
- L4: "using synthetic satellite images": Please give more information how these images are generated in the abstract since this is part of this study.
- L6: Success rate of what?
- L7: Which cities did you choose? Please elaborate a bit.
- L8: What do you mean by "error"? What are you comparing with?
- L8-10: The sentence starting with "Our learning method [...]" is clear: You already say that you reduce the error in the previous sentence. So I suggest to remove this sentence or rephrase.
- L12: Please define IQR
- L16: define UNFCC
- L16: The citation has errors. Please consider putting the author in additional curved brackets.
- L25: define OCO
- L28: please define ppm
- L30-33: Please add a citation for OCO-3 SAMs to this sentence.
- L38: define CO2M
- L38: define GOSAT-GW
- L41: Please define WRF abbreviation.
- L46: "Attempt" could imply that these studies were not scientifically justified. Please rephrase.
- L50: Sentinel 5-P does not measure $CO_2$. Please remove it.
- L50: Please add a citation for GOSAT-2
- L51: define OLAM

- L53: I'm not sure what you mean by "variability of the local background signal". Within the surrounding of a small area like a city, the natural signal should be the same everywhere which is used for verification of the satellite instruments by small area approximation e.g. described by Taylor et al., AMT, (2023) and references therein. Please explain what you mean by that.

- L57: Please remove "extensively" because this is a subjective rating.

- L58: Please define in more detail what you mean by "pseudo-image".

- L65: Please define IQR

- L76: I don't see any discussion of the uncertainty in the meteorology in this manuscript. Please explain.

- L80-81: Please elaborate which configurations in the framework of Paris you are talking about. It seems to me that you do not rely on the same values. Please clarify and refine your definition of configurations here.

- L87: Do you mean "evaluate" or "judge"?

- L88-89: If you want to mention the subsections here you should mention all subsections (Section 2.1 is missing). But I would prefer introducing the subsections at the very start of Section 2.

- L90: The configurations are not "recalled" in Sect. 3. There's just always a reference to the non-published paper. So, please provide a comprehensive summary of your study somewhere in this paper, summarizing all results needed to understand it.

- L91: Which "discrimination" do you mean? Please clarify.

- L99-105: It is not clear to me from this description how the model exactly is setup: My understanding from this is that the grid box sizes in this model are flexible and can be adapted for a region of interest, while being fully coupled to their neighbours. On the other hand, it seems as if the parameterizations are also adapted for each grid box and I'm wondering how the authors can ensure mass conservation and general consistency in this model, e.g. at which grid size do you decide to switch from hydrostatic to non-hydrostatic mode? And is the time step the same everywhere in the model, and if yes, how do you deal with the fact of wind speeds leading to motion across more than one grid box, especially in the higher resolution? A bit more details would be beneficial here.

- L96-105: In addition, you describe below that you simulate more than one month with the model. How do you achieve realistic meteorology in your model? Is it nudged towards an external dataset? Is it free-running?

- around L106: What is the top altitude of your model? Does it account for troposphere, stratosphere or only the boundary layer? As satellites measure the total column $XCO_2$ this is an important information to be added. In addition, how many model levels does your model have and how well-represented is the lower atmosphere in the vertical, i.e. how many model levels are in the boundary layer and what is the vertical grid spacing there?

- L106-108: This is not correct. There are models that use altitude and surface following coordinates successfully. So please delete this sentence or at least rephrase.

- L108-109: I would prefer either to delete these general sentences about the model or move them above to the introduction or the general description of OLAM above in this subsection. Otherwise, it is confusing with respect to the actual setup of the model in your study.

- L109: What do you mean by "It [...] allows to reduce the representation of urban plumes"? Do you mean that you can simulate the plumes more accurately using your approach?

- L110: Comparing the mentioned mesh size of 3 km with Figure 1 and the statements at the beginning of Sect. 2.3, it seems as if there is one value per hexagon, meaning that the grid points represent the hexagons with a side length of 3 km (according to Sect. 2.3) which does

not mean that the mesh size is in this order. A more readable quantity would maybe an effective side length of a rectangle with the same area as the hexagon.

- Sentence in L107-108: Does that mean that you use higher resolution at all coastlines and mountains on the globe in your configuration? Please clarify.
- L115: define ODIAC abbreviation
- L120: Which method is used to interpolate the results to the regular grid and which target resolution is used? Does the interpolation have an influence on the results?
- Figure 1: Where do the large $XCO_2$ values in the very south of the left panel come from?
- Figure 1: It would be helpful if you could include an example where the pseudo-images are located in this figure. In addition, "twice the size" is wrong because it relates to the square edges in both dimensions, therefore it should be "four times".
- L122: It seems from Fig. 1 that the mentioned regular grid is highly oversampling the native grid, but this should be noted somewhere here. In addition, it is not clear from the description whether $XCO_2$ is calculated on the native grid and then interpolated or vice versa. Figure 1 suggests calculation on the native grid and then interpolation, but the text describes it the other way. Please clarify.
- L125: The resolution should be moved upwards, including information how the data are interpolated. See comment above at L120.
- L128: As mentioned above, if there's one value per hexagon, the area of the hexagon is most relevant here, which is 23.4 $km^2$ for the model, 3.1 $km^2$ for OCO-3 SAMs and 6 $km^2$ for CO2M. Therefore, the hexagons at the highest resolution are still 4 to 7 times larger than the satellite footprints.
- L129-130: "[...] on which the analysis will be conducted" What is the mentioned spatial grid? Is it the actual satellite footprints?
- L129: It is not clear which "simulated patterns" you are talking about here. Please clarify.
- L130: Please provide the date notation in the standard format defined by Copernicus. In addition, these are 41 days, but obviously only 40 days are used to get the 9920 images. Please clarify.
- L132: Please clarify where the number of 9920 images comes from, because 8 hours per day x 41 days x 31 cities does not arrive at this number.
- L135: "that expected for CO2M" - Please mention the swath width of CO2M somewhere in the paper (maybe best in the introduction).
- Section 2.3: I understand from your description that you interpolate your simulation results to a 1x1 km grid and use this as a proxy for what the satellite sees. It would be much better if you used real orbit data from both satellites and account for the actual footprints of them (which you do anyway in your preprint Danjou et al., 20xx).
- L144-157: This procedure seems very complicated to me and also for the application to real satellite measurements. It would be great to have an illustration of the distribution of emission targets in a city and what is the benefit with respect to a simple circle of a specific size around the city center. I am sure there are reasons for you to do this procedure but it is not clear to me what these reasons are from the description in the text.
- L164: If the metropolitan areas from GRUMP are larger than a SAM what is your argument to decrease their size in your study? If the only reason is that you decrease it so you can study it, this would be a recursive argumentation and not scientific.
- L165: I am sure you can redefine these metropolitan areas in the model to your definition (why should it not be possible?). By that, this analysis would become feasible and possible.
- Sect. 3: Instead of always mentioning the non-published manuscript and comparing with it, it would be much better to describe your methodology here completely, so that the reader can

follow your steps (or as mentioned in the general comments provide a comprehensive summary of the other study at some point and refer to that).

- L170: Also here, an illustration would be helpful. For instance, does this plume boundary account for the fact that the emission zone is an extended zone or does it just start in the center of the image?
- L171: "over the entire image": For point sources such as power plants, only the wind at the location of the emitting target is usually used to determine the direction of the plume (e.g. Nassar et al., 2022). Although this is not a point source, I think the average should be taken for the emission zone, only.
- L175: I think you mean an estimate of the enhancement in the plume.
- L176: Is the 5 degree polynomial used because of changes in the wind direction? Can you estimate the emission when the wind is so variable? There will be also mixing into the plume when the wind varies in its direction which is why usually only times are used where the wind can be assumed to be uniform.
- L182: "averaged wind" By that you already assume that the wind is uniform all over the analysis area and the PBL. Therefore, I do not understand the 5-degree polynomial mentioned earlier.
- L186: Is W the horizontal wind?
- L187: The value in brackets of sigma should be x.
- L187: "the mass of CO2 in the atm. column per unit area" I assume you mean the enhancement in the plume?
- L190: It is very confusing to talk about "modelled" here, because it could also mean some $XCO_2$ modelled by your OLAM simulations. Please rephrase and mention that it's the Gaussian plume model you're using here.
- L191: Similarly to the previous comment: "Observed" is very confusing because you're using simulated values everywhere. Please rephrase.
- L192: Delta $XCO_2$ in the equation should be dependent on x and y
- L192: As this equation seems to be a numerical value equation units should be given to the quantities. What is Delta XCO2 here exactly? And I assume the M's are molar masses?
- L193: I would call the surface pressure P_{s,dry air}(x,y) because it depends on the x and y direction.
- L194: The description would be much clearer if you would mention that r, a and F are free parameters in Eq. 1 which you want to fit here to get the best estimate in terms of the Gaussian plume model.
- L197: Where does the "average" radius of the city come from?
- L201: Why is the normalisation needed? And why are you using exactly this normalisation? Please add this information here.
- L202: "for clarity": Why did you choose these limits? Please explain.
- L203: The angle should be "Theta" and not "Theta_init" and the "Element" symbol is missing.
- L204: "defined as optimal by Danjou et al. (20xx)". Please provide a summary of the non-published manuscript somewhere in the paper with all the results needed to understand this study.
- around L210: The description would be much clearer if you would mention that the actual shape of Paris is nearly circular whereas this is not generally the case for all cities, which is why you had to adjust the radius of interest.
- L211: What does "4.3" mean? Please clarify.

- L215: I do not understand what you mean by "directly above the city". From my understanding, you have to estimate the emission downwind of the emission target. Please clarify.
- L218: As the IME abbreviation is not in the main text, it would be better if you defined it in the Appendix C where it is actually used.
- L221: appendix --> Appendix C (or actually Appendix A if the Appendices A and B are removed as suggested above)
- L222: I think you would like to do an error reduction of your analysis, so please rephrase the title accordingly
- L222: Please explain first what you mean by the "sensitvities" and what is the general purpose of this section before going into the details.
- L229: You have not mentioned how the error on the emission estimate is calculated before. Please add this information to the previous section about the methodology.
- L231: As mentioned above, it is not clear from your description why you are doing this sensitivity analysis. A motivation for this is needed.
- L233-234: "a way to define ...": This is the motivation. Please move this to the front of the section.
- Sect. 4.1: I think you're saying here that you are binning your images according to percentile thresholds in 5% steps. But it is not clear from your description if you do this for each city separately.
- L263: "variance reduction": So your error mentioned earlier is the "variance"? Please explain.
- L266: "the depth is set to 2": You do not mention here, which criteria are used in the end, which is confusing. It would be better to combine this discussion with Sections 5.2.1 and 5.2.2 where the criteria are discussed, also because the second choice ("diagnostic" variables) depends on the first choice.
- Sect. 4.2.2: It seems as if your decision tree is a two-step procedure: First, you do it for the predictable variables, then you do it for the remaining images using the diagnostic variables. Please include this information somewhere.
- L285-286: The sum of the number of variables in brackets do not match the ten diagnostic variables.
- L286: I think it would be appropriate to put the table here as part of the main text. Otherwise, the reader has to go to the end of the paper to understand what you are referring to here.
- L290: There are many peer-reviewed publications highlighting the importance of wind in the calculation of emissions from emission targets, so please use another publication here.
- L294-298: So your suggestion is to apply exactly this method with the same thresholds to real satellite data for these cities?
- L299: You have already characterised the background $XCO_2$ in the sections before. Do you mean you want to characterise the variability of the background because this could lead to errors?
- L300: It is clear that the background is crucial for the analysis of the emission plume. There are many publications highlighting this, so please use another citation for this.
- L303-304: It is your choice to define a fixed size of the images. You could easily include a variability in the size of your images to analyse this effect, e.g. by adding a random parameter to the edge size of the images. Because you're saying this could be important this would be worth doing.
- L314: This is the very first mentioning of GP2. Please define what you mean by that.

- L314: It is unclear from the description where this number of 92% comes from. Please extend your explanation.
- L315-316: Does that mean that the uncertainty of your method is 78% which would be huge and mean that we basically cannot infer much information from that? Please clarify. An illustration of this would be helpful, too.
- L320: "the variability of the error distribution remains large across cities": Again, an illustration would be very helpful here to understand what is meant by error distribution.
- L324: Are you saying here that your method can only applied to single cities?
- L331: "significant": Did you check for statistical significance here? If not, please replace by "increased".
- Fig. 3 caption last line: left-hand --> right-hand
- L335: Since Appendix C has many subsections, please refer to the correct subsection here.
- L337: "impairing our ability to determine the optimal set of thresholds": If the variables are correlated they do not provide additional information, so you should choose independent variables.
- Section 5.2: These results should be moved to a much earlier place, because otherwise the manuscript is hard to follow.
- L352-357: I think what you want to say here is that you calculate the median threshold and remove all data that are beyond the threshold where the error increases. Please clarify.
- L354-355 and L356-357: Why is it important how many images are between the bounds? I do not see the importance of these sentences.
- Fig. 5 caption: It would be good to add the absolute emissions of the 31 cities you investigated at some point of the manuscript. Do they differ by many orders of magnitude?
- Fig. 5: From the description in the main text, I think the values on the y-axis are not given in percent but in ratios to the emission.
- L362: Since it is not explained at all in this study up to now, where the "error" comes from, this statement cannot be validated. Please explain in detail at some point how you define and calculate the error in your analysis.
- L424: Please repeat these here or somewhere else in the paper since your previous study has not been published yet.
- Fig. 6: Please add numbering to the panels and refer to them in the text. Otherwise, it is not clear which panel you are referring to.
- L448: The orbit of OCO-3 is not really predictable so that the overpass can happen at any local time during daytime. Please clarify.
- L449: Why are you using 11° here and not 12°?
- L451: It is clear that a lower resolution will result in smoothed and more homogeneous wind speeds/directions.
- L452: As mentioned above, the resolution of your model is not 3 km but rather something around 5 km.
- L453: "are located in Asia and America": Which is surprising because these cities have supposedly the highest emissions. How do you explain this?
- L456-460: This should be mentioned much earlier. Please mention at your earliest convenience that you are interested in clear-sky conditions only here.
- L461-462: "as current instruments cannot make measurements mover both water and land in a limited time and space interval.": I think it would be better to say that the signal-to-noise-ratio is lower over water making measurements more challenging.
- L463: I thought it was 20 km. Why is it 30 km here??

- L465-468: I think this should be part of your methodology and not of your discussion of the results because this makes the connection between your model simulations and the real satellite measurements.
- L468: Why 11 degrees? Above, you derived a threshold of 12 degrees.
- L471: As can be seen for the cities at the west coast of middle Africa, there are regions with large cloud frequency in Africa. Please rephrase.
- L474: "no more observable cities": Please rephrase because this means that there are no cities that can be observed. You could e.g. write something like "the number of cities does not increase..."
- L476: "if observed daily": maybe better "if there were daily SAMs or overpasses"
- L477: Please move this sentence above to around L466 where you describe the general procedure.
- L480 and L484: "stand[s] out" is not a scientific notion. Please remove and just write the facts, e.g. "In Australia, only five cities..."
- L481: "For this contintent": Maybe better talk about the five cities instead of the contintent.
- L484-488: I think it would be much better here to talk about absolute number of cites instead of the relative number since the number varies a lot between contintents (e.g. 5 in Australia and 273 in Asia).
- L496: "xx vertical levels" Please include the number. In addition, the vertical resolution in the boundary layer is of relevance here.
- L497: Do you mean 137 levels?
- L500: "is less sensitive to sampling": I do not understand what you mean by that. For a given grid spacing, you get variability only for a certain resolution for all variables. Please clarify.
- L518: "precise" What do you mean by precise? How representative are these thresholds for real applications?
- L532: But one of the strengths of your method is that you can give some indication which cities can be used for satellite observations. So, which cities come out of your study whose emissions can be observed from space? This conclusion is missing here.
- Code and data availability: I don't think that this is conform to AMT policy. Please add your data to a repository to be publicly available.
- Appendices A and B: Actually, I do not see the reason to put these figures into the appendix of the paper. They can be part of the main text. Their discussion is done in the main text only anyway at the moment.
- Fig. A1: Please add information to the caption how the emissions shown in the panels are calculated.
- Fig. A1 caption: "OLAM boundaries": Please clarify: Is this your estimated area of the city or is this the GRUMP product mentioned in Sect. 2.4? If it's the GRUMP product, I would prefer that you show the boundary of the city you use in your analysis.
- Table B1: As mentioned earlier, I'm missing the cloudiness as a parameter here which will be very important to select times suitable for satellite measurements.
- L547: "in over 98%" Don't these methods vary in this number?
- Fig. C1: The quantile for the emission budget is not in the range between 0 and 1.
- Fig. C1: The maximum true ratio in panel (e) is 5 whereas it is 3 for the retrieved ratio in panel (d). How can you explain these differences?
- Fig. C1: panel (f) please add units of the standard deviation
- Fig. C1 panel g: Where does the local minimum at the 0.5 quantile in both methods come from?
- Fig. C1 caption: Please add to the caption why there are only 2 lines in panel (g).

- Caption of Fig. C1: "error" with respect to what?
- L567-568: This sentence "Cities with emissions..." does not make sense to me. Please rephrase.
- L571: Please remove "very"
- L572: Please add more information at which quantiles they are similar.
- L573: Where does the "real" anthropogenic signal come from?
- L578: Please remove "very" or define its dependence e.g. by numbers.
- L583-586: I don't understand these arguments. From my understanding, if the variables are correlated, only one of them should be included in this analysis in the first place.
- L590: There is no section 5.3.4.2
- L590: There is no table 5.2.
- L592: Why is the number lower for CS and IME?
- L593: Remove "very" here.
- L598: Again, there is no table 5.2. Please update.
- L608 and L609: I do not understand what you mean by "standing out". Please rephrase and explain.
- L611: Why do you now use the same thresholds as for GP2? The thresholds will depend on the method you use. In the previous section, you showed that the thresholds are different for each method and to use the optimal one for each method would be the way to go.
- L611: The description in section 5.2.1 has never been referred to as GP2 method. Please add this to this section.
- Fig. C2: Please add the unit to the y-axis
- Fig. C2 caption: Do you mean emissions > 2.1 ktCO2/h?
- L614: What do you mean by "accuracy"? Is it the median or the spread or both?

**Technical corrections:**

- L28: Please use \citep instead of \citet
- L29, L37, L38: XCO2: subscript for number 2
- L30: \citep instead of \citet
- L59: remove second "the"
- L61: "is" --> are
- L83: help to identify
- L84: help to identify
- L88: Remove "The" at the beginning of the line.
- L88: move "used" before OLAM simulations
- L97: Use \citep for Ullrich citation
- L116: I think, it should be "power plants"
- L145: a --> an
- L154: 2063 km2 (squared is missing)
- L156: Remove comma between "Note that" and "the"
- L161: Use \citet for the citation
- L163: citation is wrong
- L201: It would be better to include commas between the vector elements.
- L214: annex --> Appendix
- Figs. 2 and 3: Please switch the order of the figures to match the mentioning in the text.
- Sect. 4.2.1: Please convert the verbs from future to present, e.g. "will separate" --> separates

- Fig. 2 caption: will classified --> will be classified
- L294: "are based" --> is based
- Fig. 4: black line --> black dotted line
- Fig. 4 caption: the the --> the
- L397: modelisation --> simulation
- Fig. 5: typo in y-axis caption: % of the true emissions
- Fig. 6: right panel x-axis: Better "meeting" instead of "combining".
- L476: "an order of magnitude of" --> approximately
- L484: "of cities of" --> of cities with
- L504: "5.2.2" --> "Sect. 5.2.2"
- L511: "XCO2" add subscript
- Fig. A1: Please add more space for the Ningbo panel.
- Fig. A1: Please reverse the color scale because it is confusing that red means lowest emissions.
- Caption of Fig. A1: Add "Sect." to 2.3
- Caption of Table B1: 1first --> first
- Caption Table B1: W2D --> W (which actually occurs in the table)
- L538: "optimized" --> "applied" or "investigated"
- L539: "across" --> for
- Caption of Fig. C1: remove "according"
- L568: Remove "emissions." at the end
- L598: close --> similar

---

## Referee Comment (RC2)

The manuscript addresses an important scientific issue and presents an innovative approach to assess CO2 emissions from urban areas. However, many parts in the preprint suggest that it is work in progress. The study holds potential but necessitates substantial revisions to strengthen its clarity as well as scientific rigor.

**General comments:**

The manuscript should make a clear distinction that its primary objective is not the inversion process itself, but rather the assessment of the confidence in the inversion results via predictable and diagnostic variables.

In the methodology section, more emphasis could be put on how the two-step procedure via predictable and diagnostic variables contributes to the accuracy assessment of emission estimates.

The manuscript heavily relies on Danjou 20xx, which is not accessible for verification. I think this is problematic and impedes a thorough evaluation in parts of this study.

The manuscript lacks clarity on what constitutes the error. The reference data used for determining the error is not explicitly defined.

The manuscript does not provide validation of its results with actual satellite data (e.g., OCO-3 SAA) or ground truth measurements. The absence of this validation makes the reliability of the simulated results vague to some extent.

The variability of the error distribution remains significant across different cities. What are the implications for estimations from real satellite images?

The study should discuss the detection limits of current and future satellite missions and how those might impact the results.

Is the purely random noise model imposed on XCO2 data in the study representative of real-world atmospheric and environmental conditions? In actual scenarios, factors like surface reflectivity of different land types and the presence of aerosols can introduce more structured or systematic errors rather than purely random ones. Would incorporating more realistic, structured errors enhance the model's applicability and accuracy in real-world urban CO2 monitoring scenarios? Given this potential limitation, what implications does this assumption have for future research?

The selection criteria for the size of the target emission zone radius, are not comprehensively described. I can't find a clear rationale for the chosen size of the emission zone radius. Any potential to estimate this radius through an inversion approach?

The influence of cloud coverage on satellite observations is a significant factor that the manuscript should address.

The authors of the manuscript should revise Section 3 to summarize only the key aspects of the Danjou 20xx study that are directly relevant to their current research. As mentioned above, it is problematic that the Danjou 20xx paper is not yet available and that the study heavily relies on it.

The description of the OLAM model and its simulations should be more comprehensive in order to better understand the simulation results.

The manuscript acknowledges that the method is not universally applicable to all cities and is still a work in progress. However, it should also describe how the proposed method would be applied to real satellite measurements and critically assess, whether its applicable at all within the error budget.

The conclusion should reiterate the study's focus on developing a procedure for accuracy estimation, summarizing how the study advances this goal and its implications for future research and practical applications.

**Specific comments**

The manuscript frequently uses abbreviations and technical terms without defining them (e.g., $XCO_2$, UNFCC, OCO, ppm, GOSAT-GW, WRF, OLAM, IQR).

3: The phrase "selecting images to be processed" should be more clearly defined.

Fig 1: Identifying the factors behind the peak $XCO_2$ values in the simulation domain deyond the city boundaries.

46: The Danjou et al. (20xx) should be made available to the reviewers due to its significant relevance to the research.

53: Clarify local background signal (not clear to me, what is meant by this term).

56, 58: "This study" can be misleading. Write "The study".

48-50: Consider moving the sentence up to line 25, before you start describing the OCOs.

261: Set by the user? Do all produce the same value?

405-406: Please clarify why Paris exhibits such a low emission rate? Is this due to the absence of significant point sources?

418: What are the implications or consequences for the study if the dependencies of errors are not completely comprehended, even when using synthetic data.

Fig. 5: Typo in y-label (thrue).

316: The GP2 inversion method is presumed to be a variation of the Gauss Plume approach. However, at this point it is unclear for me what the '2' in 'GP2' represents. Please provide further details or clarification.

520: Was the 12° threshold for wind variability found empirically or is there a rationale behind choosing it? Is this higher variability threshold, compared to what's mentioned in Danjou 20xx, a result of the increased resolution in the model?

578: The error appears to be highly sensitive to the city's radius. Could you clarify what "pseudo-image filtering" specifically entails? I don't have access to the Danjou 20xx source for reference. Additionally, referring to line 550 and following, I guess it implies filtering out scenes with variability above a certain threshold?

58: What are pseudo-images? Synthetic 2D CO2 concentration images, I guess?

229: What is the primary source of error in the emission estimates? Does it originate from the tree model or the GP2 inversion method?

47: Improve the English language in this sentence. It encapsulates the primary motivation of the study and thus should be distinctly emphasized and articulated.

76: The frequent citation of Danjou 20xx for all the "light" methods seems inappropriate. It would be more suitable to refer to the original papers specific to each approach.

74-79: The logic presented in this paragraph is unclear. It would be beneficial to revise and clarify the content for better understanding.

Sec. 2: Mention that ECMWF ERA-5 product is used for meteorological data. It is only mentioned in line 446.

130: What is the methodological rationale behind assuming constant emission rates? Do you expect that incorporating dynamic emission rates would significantly alter the study's results or conclusions?

217-221: Does the observation that all methods yield similar results suggest that the assumptions in the model used to generate the synthetic data might be overly simplistic?

Sec. 3: Does this paragraph solely describe the work done in Danjou 20xx? Certain statements, like in line 160 "The method used here...", are ambiguous in the current context. Does "here" refer to this study or to Danjou 20xx? It may be beneficial to condense the paragraph preceding Eq. (1) for clarity.

217-221: Consider clarifying this point earlier in the document, perhaps where the Gaussian plume inversion (Eq. (1)) is introduced, for better coherence and understanding.

---

## Author Comment (AC1)

**Review of "Optimal selection of satellite XCO$_2$ images over cities for urban CO$_2$ emission monitoring"**

Alexandre Danjou[1], Grégoire Broquet[1], Andrew Schuh[2], François-Marie Bréon[1], and Thomas Lauvaux[1,3]

[1]Laboratoire des Sciences du Climat et de l'Environnement (LSCE), IPSL, CEA-CNRS-UVSQ, 91191 Gif sur Yvette, France
[2]Cooperative Institute for Research in the Atmosphere (CIRA), Colorado State University, Fort Collins, USA
[3]Molecular and Atmospheric Spectrometry Group (GSMA) – UMR 7331, University of Reims Champagne Ardenne, 51687 Reims, France

**Correspondence:** Alexandre Danjou (alexandre.danjou@lsce.ipsl.fr)

*We would like to thank both reviewers and the editor for the careful and detailed reviews and their patience during this process.*

*It seems to us while reading the reviews that the aim of the study and of the different analysis conducted were not sufficiently explained, which perturbed the reviewers and undermined the comprehension of the links between the different parts. We thus have made numerous changes on the introduction and took time to better introduce each sections. We have also changed the title as we found that it might give the impression that we used a model for our inversions, whereas we only use our model to generate our synthetic images. The description of the learning method has also been completely rewritten to make it clearer.*

*Multiple comments were in relation to the article Danjou et al. 2024. This article is now published and accessible (https://doi.org/10.1016/j.rse.2023.113900). Some of the references to it were unecessary and have been removed. We also tried as much as possible to gather the references and explain them when necessary. We hope that the article is now more self sufficient in regard to Danjou et al 2024.*

**1 Review 1**

In this manuscript, the authors discuss a methodology to estimate CO$_2$ emissions by cities in an automatic way to validate self-reported emissions by cities. For this, they use CO$_2$ concentrations simulated by a global model with an adaptive horizontal resolution enabling them to increase the resolution to about 23 km$^2$ locally around 31 cities with improved representation of the meteorology and potentially better comparability to satellite measurements around these targets. Their methodology consists of calculating hourly XCO$_2$ during local daytime from their simulation in a 150x150 km$^2$ region around the city and then derive criteria to select images which they conclude will be appropriate for satellites to observe the emission plume using instruments like OCO-3 or CO2M. While it is an important scientific topic the authors try to tackle, the descriptions in the manuscript are confusing and have many imprecisions making the manuscript hard to follow, which is why I suggest major revisions before the manuscript should be even thought of being published. In addition, it seems that the manuscript has been submitted in a

preliminary stage because almost no abbreviations have been defined, some values are "xx" (e.g., L496) and there are many typos in the figures. My comments are separated in general and specific comments, followed by technical corrections.

*We'd like to thank the reviewer for his detailed comments. We apologize for the typos and other mistakes, which have been left during submission. We acknowledge that these highlight a need for a better proofreading of our manuscript. We have now carefully reviewed it which has led to a significant improvement of the whole document.*

**1.1 General comments:**

– Citation Danjou et al. (2024): While it is appreciated that it is made clear by the year number 20xx that it is a preprint under review, the whole manuscript is based on this non-published study (cited 40 times). In order to make the results of this current manuscript understandable for the general public, it is important to add a comprehensive overview with all the results from your previous study needed to understand the results in this manuscript. In addition, please add the journal the manuscript has been submitted to so that at least it may be found in the future. You could also consider to publish the preprint in a citable space and cite this here, which would make this manuscript here much more transparent at this stage.

*There may have been a problem in the process of sharing the files with the reviewers (we apologize for this failure). We actually sent the manuscript of Danjou et al. (2024) to the reviewers to support the review. As planned at that time, the paper Danjou et al. (2024) had been accepted in RSE and is now published.*

– It is never defined which is your reference determining the "error" which is mentioned at many points in the manuscript (e.g., L7, L91, L362 and many others). Unless the reference data of the city emission is not clear, the whole error analysis does not make sense, so please describe clearly what you use as a reference at some point of the manuscript. See also specific comment for L362. In addition, this also means that the motivation of this study is not clear: Are you investigating how well suited different methods are to determine emissions by cities (model as "true" emissions and method as uncertainty) or are you interested in which cities and meteorological conditions are best for analyzing city emissions with satellites, which is suggested by the title but where it is unclear what is your reference?

*The first proposition of the reviewer corresponds to the aim of the article Danjou et al. 2024. Here, as he precisely pointed out in the second proposition, we are interested in which cities and meteorological conditions are best for analyzing city emissions with satellites. We have replaced in the forth paragraph each occurrence of "this study" by "their study", when it was designating Danjou et al. 2024, as it must have been part of the confusion. We have also rewritten the 6th paragraph of the introduction and included a definition of the error : "As we are working with synthetic data, the error in the emissions estimate is directly accessible by comparing the emissions estimated by the inversion method with the synthetic true emissions used in the OLAM simulations."*

*The 6th paragraph is now : "The objective of our study is to resume the series of analysis from Wang et al. (2018); Schuh et al. (2021); Danjou et al. (2024) and deepen the evaluation of the conditions corresponding to reliable estimates of urban CO2 emissions using satellite XCO2 images. To do this, we use a little more than a month of simulations of*

55 *local XCO2 scenes over large cities. This simulations are generated with the global OLAM model and evaluated by Schuh et al. (2021). We use these simulations to generate synthetic satellite images for the selected cities, and estimate their emissions by applying one of the automated and computationally-light inversion methods implemented, tested and optimized by Danjou et al. (2024). By using realistic simulations to derive the synthetic image and using a method independent of the model used for the simulations to estimate the emissions, we take into account realistically the*

60 *uncertainty in the meteorology, atmospheric transport and background. As we are working with synthetic data, the error in the emissions estimate is directly accessible by comparing the emissions estimated by the inversion method with the synthetic true emissions used in the OLAM simulations. The study of the emission estimation error for different cities and weather conditions aim to support the identification of criteria for discriminating between images, separating those whose processing yields statistically reliable estimates from those whose processing is statistically unreliable."*

65   – A major issue with satellite observations are clouds which will decrease the coverage. They are not mentioned in the methodology description (Sect. 4) in any way, but will probably be the main limitation to the emission estimates. I'm sure you can derive cloud information from your simulations which in my opinion has to be the first criterion to select whether emissions can be estimated using satellite measurements. This cloud screening would be something like a pre-selection of the images. Otherwise, the selection of the images is not "optimal" as suggested by the title.

70 *When it comes to determining which cities are most suitable for measurement, cloud cover is indeed a major issue, and the frequency of cloud cover is an important criterion for city selection. We examined this point in section 6.2.*

*We have reserved this analysis for discussion because realistically adding the effect of clouds in images (filtered pixels and contamination of neighbouring pixels) and trying to objectively quantify their impact on error is of a complexity that would merit a dedicated study to do it more properly than what is proposed in the discussion.*

75 *We also believe that, once an image is partly contaminated by clouds, it is not worth processing. Indeed, the effect of cloud presence on the measurement is very complex, due to the presence of 3D cloud radiative effects in OCO-2 retrievals (https://doi.org/10.5194/amt-14-1475-2021). Given the size of these effects (of the order of a few kilometers) and the size of our images (a few tens of kilometers), we doubt the value of trying to obtain an estimate from an image contaminated by clouds. However, this remains to be demonstrated, and the complexity of such a task requires, in our*

80 *opinion, a complete study and has no place here.*

*Adding clouds would not alter the conclusions we have reached, the criteria identified should still be important even in the presence of clouds. Thus, adding cloud coverage would only increase the error and add another criterion.*

  – It seems as if it is not accounted for detection limits of the satellites at any point in the manuscript and should be considered or at least mentioned somewhere. Can a satellite like OCO-3 or CO2M even distinguish emissions in the

85     order of 2.1kt/h from the background?

*The term detection limit is often used in the context of the detection of unknown sources (methane super-emitters, for example, such as in Lauvaux et al, 2022). Here the problem is different: we already know the location of the sources (we*

*know where the cities are) so we don't have to detect them but "just" to quantify their emissions. The term quantification limits is therefore more relevant. It is in fact the whole point of the article : to determine what conditions make it possible to quantify a city's emissions by satellite. We show that the city's emissions is one of the two most important criteria and we quantify a threshold on it and the error associated.*

- All the results are based on simulations by the OLAM model, but the description of the model and of the performed simulations are not comprehensive, many things are missing making it hard to understand, see specific comments below for lines 95 to 138 and for Fig. 1.

*See our answers to the cited specific comments. We have also took care to more broadly revise this section and hope it is more understandable now.*

- You are mentioning at various points in the manuscript that you are interpolating and extrapolating without including information which methods are used and why you are using them. Do these interpolations to a 1x1 km grid influence the simulated emission fluxes? Are they mass-conserving? Why and where are you extrapolating the $XCO_2$ values (L173)? Common practice is to use the average of the background region. What is the reason that you use another procedure?

*Interpolation (l. 125 and 132) concerns the creation of images with 1km by 1km square pixels from data supplied by the OLAM model on a hexagonal grid. This interpolation is classic and conserves mass. It's all part of the synthetic-image generation process and not the inversion method.*

*Concerning the extrapolation, Danjou et al. 2024 shows that using the mean to estimate the background is a very inaccurate estimator, hence the choice of a more complex extrapolation. We won't go into more detail here, as all this is described in detail in Danjou et al. 2024 (now available).*

- I am missing a description how you suggest to use your method for real measurements. How do you suggest to derive the variables needed for the analysis with your method with respect to real satellite measurements?

*Add to lign 295 : "When using real satellite images (which is out of scope of this study), meteorological variables can be derived from weather products such as ERA-5 (Hersbach et al., 2018). City characteristics can, as in this study, be calculated from gridded inventories sur as ODIAC, and from database on urban land cover and population/socio-economic activities such as GRUMP."*

- I would prefer reordering the methodology section because it is hard to follow in the current setup. First, I suggest putting the figure and table from Appendices A and B (Fig. A1 and Table B1) to their place in the main text where they are discussed as they have not any description in the appendices anyway. Second, it would be much better to move the descriptions from Sect. 5.2 to the decision criteria selection part in Sect. 4.2 and move the list of variables before this. This would improve the readability because otherwise, the reader is left with the methodology without the outcome in the current Sect. 4.2. *We have moved the table in the article. Concerning the figure, we think that its interest is merely illustrative and that the space it would take up in the main body is far too large compared to its interest.*

120  *Concerning the reorganisation of the sections, most scientific articles present the method and the results in a separate way. Here we are following the usual order and think that mixing the two would destabilise the readers. We have also made it clear that parts 4.1 and 5.1 go hand in hand and form a preliminary analysis and rewrote the introduction of the sections to make the links and justification of the different sections clearer.*

– At some points you basically say that this is still work in progress and that your method is not applicable to all cities
125    (L320, L324, L527) because it may depend on the surroundings of each individual city, which is okay but then the abstract should provide this information. *We have added at the end of abstract : "Despite this efficient filtering, the accuracy of the estimates varies widely from city to city within the group identified as the most accurate."*

**1.2 Specific comments:**

L1: define XCO$_2$ *changed to "CO$_2$ column-average dry air mole fraction (XCO$_2$)".*

130   L4: "using synthetic satellite images": Please give more information how these images are generated in the abstract since this is part of this study. *Addition of the abstract of : "It uses synthetic data experiments with synthetic truth and 9920 synthetic satellite images of XCO2 over 31 of the largest cities across the world generated with the Ocean Land Atmospheric Model (OLAM) zoomed over these cities. We use a decision tree learning method applied to this ensemble of synthetic images to define criteria on these emission and atmospheric condition for the selection of satellite images."*

135   L6: Success rate of what? *Replace "has a success rate of 92%" by "manages to produce estimates for 92% of the images"*

L7: Which cities did you choose? Please elaborate a bit. *replace "cities worldwide" by "of the largest cities across the world."*

L8: What do you mean by "error"? What are you comparing with? *Error on the emission estimate. The answer to the following comments makes this comment obsolete.*

L8-10: The sentence starting with "Our learning method [...]" is clear: You already say that you reduce the error in the
140  previous sentence. So I suggest to remove this sentence or rephrase. *Changed to "Our learning method identifies two criteria, the wind direction's spatial variability and the targeted city's emission budget, that discriminate images whose processing yield accurate emission estimates from those whose processing yield large errors."*

L12: Please define IQR *Replaced "biais" by "median error" and "spread" by "InterQuartile Range".*

L16: define UNFCC *Replaced "UNFCC" by "United Nations Framework Convention on Climate Change (UNFCC)"*

145   L16: The citation has errors. Please consider putting the author in additional curved brackets. *citation corrected.*

L25: define OCO *The sentence has been rewritten and a definition of OCO has been added : "Observations of CO2 column-average dry air mole fraction (XCO2) at the scale of a few square kilometers from the two current Orbiting Carbon Observatory missions (OCO-2 and OCO-3) have paved the way for quantifying emissions from large (a few ktCO2/h) industrial (Chevallier et al., 2022; Nassar et al., 2017; Zheng et al., 2019) and urban (Lei et al., 2021; Reuter et al., 2019; Wu et al., 2018; Ye et al.,*
150  *2020) sources of CO2."*

L28: please define ppm *Add "-part per million-"*

L30-33: Please add a citation for OCO-3 SAMs to this sentence. *Add citep{Taylor2020}.*

L38: define CO2M *Definition added.*

L38: define GOSAT-GW *Definition added.*

L41: Please define WRF abbreviation. *Add "Weather Research and Forecasting (WRF) model "*

L46: "Attempt" could imply that these studies were not scientifically justified. Please rephrase. *We have used this word to show that estimating emissions under such conditions bears large errors , as the quoted studies underline. However, to avoid any misunderstanding, we have followed the reviewers' instructions and changed "attempts to quantify" to "the quantification of".*

L50: Sentinel 5-P does not measure $CO_2$. Please remove it. *Indeed, sorry for this mistake.*

L50: Please add a citation for GOSAT-2 *reference to GOSAT-2 non pertinent and removed. "This is made possible by the launch of new satellites (e.g. OCO-2 and OCO-3) measuring $XCO_2$ at kilometer resolutions with ppm accuracy."*

L51: define OLAM *"Schuh et al. (2021) use high-resolution simulations from an adaptive-mesh model, the Ocean Land Atmospheric Model (OLAM (??)),"*

L53: I'm not sure what you mean by "variability of the local background signal". Within the surrounding of a small area like a city, the natural signal should be the same everywhere which is used for verification of the satellite instruments by small area approximation e.g. described by Taylor et al., AMT, (2023) and references therein. Please explain what you mean by that. *This statement is wrong : the natural signal shows high variability in the surrounding of the cities (see figure 1 for example). The typical variations of $XCO_2$ due to biogenic or anthropogenic emission and uptake, or even to changes in the wind conditions, are of a few or a few tens of kilometers, which is much finer than the size of our images. Thus the local background is variable within the image.*

L57: Please remove "extensively" because this is a subjective rating. *Ok.*

L58: Please define in more detail what you mean by "pseudo-image". *Replaced "pseudo-images of $XCO_2$ concentration over Paris" by "simulated satellite images (i.e. synthetic images) of $XCO_2$ concentration generated with a meteorological-atmospheric transport model over Paris". We replaced every occurrences of pseudo-images by synthetic images, much more accurate term.*

L65: Please define IQR *Definition added in abstract and l.65.*

L76: I don't see any discussion of the uncertainty in the meteorology in this manuscript. Please explain. *The sentence was indeed badly formulated and misleading. We wanted to express that, contrary to some studies we do not use the same model to generate the synthetic images and perform the inversion. We have rewritten this part : "By using realistic simulations to derive the synthetic image and using a method independent of the model used for the simulations to estimate the emissions, we take into account realistically the uncertainty in the meteorology, atmospheric transport and background."*

L80-81: Please elaborate which configurations in the framework of Paris you are talking about. It seems to me that you do not rely on the same values. Please clarify and refine your definition of configurations here. *We have rewritten this paragraph (see previous comment) and move this discussion to the section 3.*

L87: Do you mean "evaluate" or "judge"? *"indicate" is indeed more suited. We have replaced "we hope to indicate" by "this analysis is expected to support the development of tools to evaluate".*

L88-89: If you want to mention the subsections here you should mention all subsections (Section 2.1 is missing). But I would prefer introducing the subsections at the very start of Section 2. *We have followed the reviewer suggestions and moved all the description of the subsections at the beginning of the corresponding section.*

L90: The configurations are not "recalled" in Sect. 3. There's just always a reference to the non-published paper. So, please provide a comprehensive summary of your study somewhere in this paper, summarizing all results needed to understand it. *Replace "The inversion methods are described in Danjou et al. 2024 in their optimal configurations and are recalled in Section 3" by "Section 3 describes the inversion method used to make the emission estimations for the main set of analysis in this study. The results with the three other automated methods described in Danjou et al. (2024) lead to similar conclusions and their analysis is thus summarized in Appendix B."*

L91: Which "discrimination" do you mean? Please clarify. *We have rewritten the introduction and the term is now introduce earlier.*

L99-105: It is not clear to me from this description how the model exactly is setup: My understanding from this is that the grid box sizes in this model are flexible and can be adapted for a region of interest, while being fully coupled to their neighbours. On the other hand, it seems as if the parameterizations are also adapted for each grid box and I'm wondering how the authors can ensure mass conservation and general consistency in this model, e.g. at which grid size do you decide to switch from hydrostatic to non-hydrostatic mode? And is the time step the same everywhere in the model, and if yes, how do you deal with the fact of wind speeds leading to motion across more than one grid box, especially in the higher resolution? A bit more details would be beneficial here. *The model is an icosahedral model, originally working in either the triangular or hexagonal dual spaces, but most recently being coded to mostly work in hex space. The mesh refinement procedure is documented in Walko2011 , and is also common place in other more well known icosahedral models such as MPAS (out of NCAR/US). Parameterizations working across spatial scales are obviously tricky and a constant research target. Currently OLAM uses a blend of the original Grell deep convection scheme with aspects of the Grell-Freitas scale aware scheme to accommodate this. Ref below explains mesh refinement procedure.*

*Walko and Avissar, A Direct Method for Constructing Refined Regions in Unstructured Conforming Triangular–Hexagonal Computational Grids: Application to OLAM, 01 Dec 2011, https://doi.org/10.1175/MWR-D-11-00021.1,3923–3937*

L96-105: In addition, you describe below that you simulate more than one month with the model. How do you achieve realistic meteorology in your model? Is it nudged towards an external dataset? Is it free-running? *It is initialized to realistic 3D met fields initially and then free running, i.e. forecast. In this sense, we get better internal consistency of tracer fields w/o the demanded jumps and so forth that would be required if you were forcing mass related variables, such as humidity and density, to external reanalysis. This is an area of active research now though since we'd like to use this model towards working with real observations at high resolution.*

around L106: What is the top altitude of your model? Does it account for troposphere, stratosphere or only the boundary layer? As satellites measure the total column $XCO_2$ this is an important information to be added. In addition, how many model levels does your model have and how well-represented is the lower atmosphere in the vertical, i.e. how many model levels are

in the boundary layer and what is the vertical grid spacing there? *The top altitude is 37,503 MASL. The model is a full general circulation model (GCM), as described in Walko et al 2008a and Walko et al 2008b. In other words, yes, it handles all of that.*

*The model edges are given below (in MASL, plus 0 MASL):*

*[1] 50.0000 101.4817 157.6048 218.2211 283.6903 354.4010*

*[7] 430.7727 513.2588 602.3487 698.5710 802.4969 914.7432*

*[13] 1035.9758 1166.9143 1308.3357 1461.0792 1626.0514 1804.2312*

*[19] 1996.6760 2206.3650 2438.7712 2695.6487 2979.5740 3293.3953*

*[25] 3640.2603 4023.6484 4447.4058 4915.7822 5433.4766 6005.6816*

*[31] 6638.1367 7337.1865 8109.8428 8963.8555 9907.7920 10951.1201*

*[37] 12104.3057 13378.9160 14787.7373 16344.8994 18066.0234 19968.3730*

*[43] 22045.2324 24258.5957 26626.7012 29160.3691 31871.1738 34698.8203*

*[49] 37502.7891*

*These are generated as 49 levels with fairly constant stretch ratios between levels to ensure optimal numerics.*

*We have added l.106 " The model has 49 vertical levels (from $0\mathrm{masl}$ to $37\mathrm{kmasl}$), twelve of them being in the first kilometre, which supports reliable simulations in the lower layers of the atmosphere, where the plumes are located."*

L106-108: This is not correct. There are models that use altitude and surface following coordinates successfully. So please delete this sentence or at least rephrase. *Our statement was maybe a bit strong, and we have moderate it by replacing "is optimal to avoid" by "helps avoiding". However we have to say that we do not agree with the reviewer comment. Every kind of model (and vertical coordinates systems) have their advantages and disadvantages. This is a strength of OLAM and the used coordinate system. If the reviewer does still not agree with our statement, we can delete this particular sentence as it is not essential to the paper.*

L108-109: I would prefer either to delete these general sentences about the model or move them above to the introduction or the general description of OLAM above in this subsection. Otherwise, it is confusing with respect to the actual setup of the model in your study. *We moved upward (l.100) those sentences as suggested and mixed then in the first paragraph on OLAM.*

L109: What do you mean by "It [...] allows to reduce the representation of urban plumes"? Do you mean that you can simulate the plumes more accurately using your approach? *This sentence is indeed unprecise and we have rephrased it : "In our case, it allows us to realistically represent the urban plumes of a large number of cities and the underlying large-scale variations in $CO_2$, while maintaining a global domain and an affordable computation time."*

L110: Comparing the mentioned mesh size of 3 km with Figure 1 and the statements at the beginning of Sect. 2.3, it seems as if there is one value per hexagon, meaning that the grid points represent the hexagons with a side length of 3 km (according to Sect. 2.3) which does not mean that the mesh size is in this order. A more readable quantity would maybe an effective side length of a rectangle with the same area as the hexagon. *We agree that this metric is misleading. We choose the area of the hexagon as more readable quantity, as suggested below.*

Sentence in L107-108: Does that mean that you use higher resolution at all coastlines and mountains on the globe in your configuration? Please clarify. *No, not necessarily. Dynamicists using a model like OLAM would apply mesh refinements in*

*areas where dynamics, e.g. winds/pressure, would be expected to change quickly, e.g. mountain circulations, sea breezes, etc. This is the MAIN use of modeling constructs like this (in general). For us, we are conditioned on looking at cities so we're interested in it for two things, (1) representation of high res emissions EVEN if the underlying meteorology doesn't change a lot across distance and (2) representation of high res emissions when the meteorology does change, such as cities near topography, e.g. Los Angeles, Lima, Bogota, etc. We have rewritten this sentences and hope that the changes resolve this comment.*

L115: define ODIAC abbreviation *Replace "ODIAC" by "Open-Data Inventory for Anthropogenic Carbon dioxide (ODIAC)"*

L120: Which method is used to interpolate the results to the regular grid and which target resolution is used? Does the interpolation have an influence on the results? *Add on l.121 "The XCO2 fields are calculated by the model on the hexahedral grid. [The results of these simulations are then projected onto a regular grid] at approximately 1km×1km resolution. [This is done] to simplify the analysis of the model outputs."*

Figure 1: Where do the large $XCO_2$ values in the very south of the left panel come from? *When looking at broader image on the same day, it seems to be a plume coming from the city at the edge of the image.*

Figure 1: It would be helpful if you could include an example where the pseudo-images are located in this figure. In addition, "twice the size" is wrong because it relates to the square edges in both dimensions, therefore it should be "four times". *Ok, location of the pseudo-image added. "twice" replaced by "three times", as the plotted image is a square of approximately 240km rather than 300km (changes made in the legend).*

L122: It seems from Fig. 1 that the mentioned regular grid is highly oversampling the native grid, but this should be noted somewhere here. In addition, it is not clear from the description whether $XCO_2$ is calculated on the native grid and then interpolated or vice versa. *The regular grid is at 1km and interpolating $9km^2$ sized hex grids, so somewhat oversampling. XCO2 is (and should always be) calculated on the native grid and then interpolated with these types of models (because of complexities w/ the unique irregular grid structure.) We added l.120 "The $XCO_2$ fields are calculated by the model on the haxahedral grid. The results of these simulations are then projected onto a regular grid at approximately $1km \times 1km$ resolution."*

Figure 1 suggests calculation on the native grid and then interpolation, but the text describes it the other way. Please clarify. *We clarified this, see answer to previous comment.*

L125: The resolution should be moved upwards, including information how the data are interpolated. See comment above at L120. *Done : answer to comment l.120 and changed "The original XCO2 fields on a variable resolution grid were interpolated to a regular grid at approximately 1km×1km resolution, which" to "The model output resolution of 1km×1km"*

L128: As mentioned above, if there's one value per hexagon, the area of the hexagon is most relevant here, which is 23.4 $km^2$ for the model, 3.1 $km^2$ for OCO-3 SAMs and 6 $km^2$ for CO2M. Therefore, the hexagons at the highest resolution are still 4 to 7 times larger than the satellite footprints. *There has been a misunderstanding on the size of the hexagons, which enforce the point of the reviewer on the need of a more readable quantity... The smallest cells have an approximate area of $9km^2$. Sentence changed to : "This resolution is finer than the finest resolution of the model's adaptive native hexagonal grid (hexagons of $\approx 9km^2$). Therefore, the variations of the model variables ($XCO_2$ field, wind field,..) have a spatial resolution which is coarser than the $1km \times 1km$ resolution of the model output grid, on which the analysis will be conducted."*

L129-130: "[...] on which the analysis will be conducted" What is the mentioned spatial grid? Is it the actual satellite footprints? *No. It is the 1km x 1km grid. Change integrated in the answer to the previous comment.*

L129: It is not clear which "simulated patterns" you are talking about here. Please clarify. *replace "pattern" by " variations of the model variables ($XCO_2$ field, wind field,..)", see previous comment.*

L130: Please provide the date notation in the standard format defined by Copernicus. In addition, these are 41 days, but obviously only 40 days are used to get the 9920 images. Please clarify. *The last day is excluded (simulation stops at 0a.m. on the 10th of September). It was effectively not clear. Dates changed to "08 August 2015 - 09 September 2015 (included)".*

L132: Please clarify where the number of 9920 images comes from, because 8 hours per day x 41 days x 31 cities does not arrive at this number. *see previous comment.*

L135: "that expected for CO2M" - Please mention the swath width of CO2M somewhere in the paper (maybe best in the introduction). *addition in the introdution of the resolution and swath width of CO2M. Removal l.127 of the parenthesis mentioning Sierk and the resolution of CO2M.*

Section 2.3: I understand from your description that you interpolate your simulation results to a 1x1 km grid and use this as a proxy for what the satellite sees. It would be much better if you used real orbit data from both satellites and account for the actual footprints of them (which you do anyway in your preprint Danjou et al., 20xx). *The aim of this study is more general than just the OCO and CO2M data. Analysis with real orbits would not fundamentally change the results, as the main discrimination criteria should remain the same. An interesting point would have been to study the influence of image size and resolution, which are certainly important discrimination criteria, but this would require a separate analysis due to the number of images required.*

L144-157: This procedure seems very complicated to me and also for the application to real satellite measurements. It would be great to have an illustration of the distribution of emission targets in a city and what is the benefit with respect to a simple circle of a specific size around the city center. I am sure there are reasons for you to do this procedure but it is not clear to me what these reasons are from the description in the text. *We have completely rewritten the introdution of this subsection to introduce the motivations more clearly : "The first task for urban emission estimation is to define the targeted emission zone. As the aim of our quantification is ultimately to help cities monitor actual emission reductions, we believe it is more interesting to think of the city as an area of significant emissions rather than in terms of administrative boundaries. As administrative boundaries rarely coincide with area of significant emissions, we need to define differently the boundaries of our targeted zone. Our definition of the targeted emission zones is based on approximate considerations regarding the size of plumes that can be detected in a SAM and on an identification of the most emitting pixels from the spatialized inventories (using a similar concept but a different and more straightforward approach compared to Wang et al. (2019)). Keeping in mind that the typical size of a SAM is 80km × 80km, we set the size of the targeted emission zone at roughly the size of a 20km radius disc. Thus, the emission zone we target occupies around 20% of a typical SAM and 6% of our synthetic images."*

L164: If the metropolitan areas from GRUMP are larger than a SAM what is your argument to decrease their size in your study? If the only reason is that you decrease it so you can study it, this would be a recursive argumentation and not scientific. *we indeed define a target size that we can measure. We don't see what the reviewer means by "recursive" and why our rationale*

*is not scientific. Defining cities boundaries is not a simple task, as the political definition rarely match the economic or emitting zones. What are we interested in : the emissions of the political entity or of the economic zone? Keeping in mind that the target is to monitor emissions in order to help decrease them, we think that the emission zone is more important. The GRUMP database*

330 *choose a definition "based on a combination of population counts (persons), settlement points, and the presence of Nighttime Lights" (https://doi.org/10.7927/H4GH9FVG). We, given the fact, that we are interested in the emissions budget, are defining the city limits based on the a priori core of the emissions.*

L165: I am sure you can redefine these metropolitan areas in the model to your definition (why should it not be possible?). By that, this analysis would become feasible and possible. *The simulations used in this study were not performed for this but*

335 *for the work done at CSU, and for example the cited article Schuh et al. 2021. We are only users of them and don't have control over them. Doing this kind of simulations is very time-consuming and was not worth it for this study.*

*Furthermore, a second problem arises with month-long simulations with global models for defining the exact limits of a plume. For simulations with a restricted domain, the plume leaves the simulation zone at one point and does not return, which means that the tracer only keeps track of emissions that took place in the previous hours, and not the previous days. With a*

340 *global model, this "finite memory" of the tracer does not exist and the emissions from the days preceding the synthetic image, which should be considered as background, are still in the tracer, which makes exact identification of the plume impossible.*

*As this paragraph perturb the reviewers and is merely a digression on the simulations, we have removed it.*

Sect. 3: Instead of always mentioning the non-published manuscript and comparing with it, it would be much better to describe your methodology here completely, so that the reader canfollow your steps (or as mentioned in the general comments

345 provide a comprehensive summary of the other study at some point and refer to that). *We have rewritten the first paragraph of this section : "The complete description of the inversion method and the details and justifications for its specific configuration and implementation can be found in Danjou et al. (2024). We make the assumption that the configurations chosen in the framework of their study remain optimal for other cities. This assumption seems justified, as the chosen methods for each steps differ from the discarded methods on objective criteria. This section only gives an overview of the different steps and*

350 *the adaptations (compared to the reference configuration from Danjou et al. (2024)) that were made in the context of this study." We also tried throughout the section to suppress the unnecessary citations of Danjou et al. 2024 and regroup as much as possible the necessary ones. We hope that it is now acceptable*

L170: Also here, an illustration would be helpful. For instance, does this plume boundary account for the fact that the emission zone is an extended zone or does it just start in the center of the image? *This plume boundary account for the fact*

355 *that the emission zone is an extended zone, as it includes "the pixels located above the city" (l.170). Illustration can be found in Danjou et al. 2024. We would rather not expand the description of the inversion method as its description is not the point of this article. We make it clearer on the introduction of the section that the inversion method comes from this previous study and hope that readers that are further interested in the method will look at this previous article.*

L171: "over the entire image": For point sources such as power plants, only the wind at the location of the emitting target is

360 usually used to determine the direction of the plume (e.g. Nassar et al., 2022). Although this is not a point source, I think the average should be taken for the emission zone, only. *Here we are interested in the direction of the plume over the city and it's*

*neighbourhood. Indeed, we could have choosen a smaller zone to estimate the direction of the wind. However, this direction is just a rough estimation that will be refined during the derivation of the centerline and during optimisation of the gaussian plume (as it is one of the free parameters). Therefore, we do not think that this point will make a difference in this study, but we keep it in mind for future work.*

L175: I think you mean an estimate of the enhancement in the plume. *Indeed. Change "plume" by "plume enhancement"*

L176: Is the 5 degree polynomial used because of changes in the wind direction? Can you estimate the emission when the wind is so variable? There will be also mixing into the plume when the wind varies in its direction which is why usually only times are used where the wind can be assumed to be uniform. *Yes. Cases where the variability is to high will be filtered according to the results of section 4. It has been already demonstrated that for wind slightly variable, we manage to have a more precise estimation with a 5 degree polynomial rather that just a line (Danjou et al. 2024 et Kuhlman2019). Discussing the inversion method is not the point of this article and thus we didn't detailed further those point in the section. However, we have put more emphasis at the beginning of the section that the method has been evaluated in a previous article and that the justification of the configuration should be found there. We hope that it is sufficient.*

L182: "averaged wind" By that you already assume that the wind is uniform all over the analysis area and the PBL. Therefore, I do not understand the 5-degree polynomial mentioned earlier. *The reason why we use average instead of value at the source is actually to account for the variability. Our inversion method require the use of a single wind speed value for the Gaussian model. The Gaussian model assumes an uniform wind, but we limit the impact of such an approximate assumption by feeding it with wind speed average over the right area.*

*The difference between using a 5-degree polynomial centerline or a linear centerline is quite small when looking at the error of the emission estimation, especially for the inversion method used here. But Danjou et al. 2024 et Kuhlmann 2019 found that it nonetheless gave better results, especially for cross-sectionnal inversion method and more complex gaussian model.*

L186: Is W the horizontal wind? *It's the effective wind, i.e. a 2D vector that simulate the average horizontal wind speed driving the plume. We have moved its definition to the l.182.*

L187: The value in brackets of sigma should be x. *Yes.*

L187: "the mass of $CO_2$ in the atm. column per unit area" I assume you mean the enhancement in the plume? *change to "$CO_2$ mass enhancement of the plume in the atm. column per unit area".*

L190: It is very confusing to talk about "modelled" here, because it could also mean some $XCO_2$ modelled by your OLAM simulations. Please rephrase and mention that it's the Gaussian plume model you're using here. *We think that the precision of $\Delta\Omega_{gp}$ makes it quite clear what we are talking about. We nonetheless changed "between the modelled mass per unit area ($\Delta\Omega_{gp}$) and the observed mass per unit area. The observed mass" to "between the mass per unit area simulated by the Gaussian model ($\Delta\Omega_{gp}$) and the pseudo-observed mass per unit area. The pseudo-observed mass"*

L191: Similarly to the previous comment: "Observed" is very confusing because you're using simulated values everywhere. Please rephrase. *We are very clear that we only use synthetic data in all this article. We do not see why the reviewer is confused. We nonetheless added quotation marks around the "observed".*

L192: Delta $XCO_2$ in the equation should be dependent on x and y. *(x,y) added.*

L192: As this equation seems to be a numerical value equation units should be given to the quantities. What is Delta XCO$_2$ here exactly? And I assume the M's are molar masses? "$\Delta\Omega(x,y) = \frac{M_{CO_2}}{M_{dry\,air}} * \Delta XCO_{2\,obs}(x,y) * 10^{-6} * \frac{P_{s,\,dry\,air}(x,y)}{g}$, where g is the Earth's gravity (in $\mathrm{m/s^2}$), $P_{s,\,dry\,air}$ is the dry air surface pressure (in $\mathrm{Pa}$), $M_{dry\,air}$ and $M_{CO_2}$ the molar mass of dry air (28.97g.mol$^{-1}$) and CO$_2$ (44.01g.mol$^{-1}$) and $\Delta XCO_{2\,obs}$ the observed plume enhancement (in ppm)."

L193: I would call the surface pressure P$_{s,dryair}$(x,y) because it depends on the x and y direction. *Done*

L194: The description would be much clearer if you would mention that r, a and F are free parameters in Eq. 1 which you want to fit here to get the best estimate in terms of the Gaussian plume model. *"are optimized during this minimization."* *replaced by "are free parameters in equation* **??** *that are optimized during this minimization."*

L197: Where does the "average" radius of the city come from? *Addition "[noted $r^{init}$]) defined as the square root of the city surface divided by $\pi$."*

L201: Why is the normalisation needed? And why are you using exactly this normalisation? Please add this information here. *Despite not being necessary, it is usual to rescale data during optimization to facilitate convergence.*

L202: "for clarity": Why did you choose these limits? Please explain. *Addition l.203 of "This bounds are fixed to avoid unrealistic results (e.g. detected plume direction perpendicular to the wind, high $CO_2$ uptake from the city,...)"*

L203: The angle should be "Theta" and not "Theta_init" and the "Element" symbol is missing. *Indeed : $\theta^{init}[\theta^{init} - \pi/4; \theta^{init} + \pi/4]$ is changed to "$\theta \in [\theta^{init} - \pi/4; \theta^{init} + \pi/4]$".*

L204: "defined as optimal by Danjou et al. (2024)". Please provide a summary of the non- published manuscript somewhere in the paper with all the results needed to understand this study. *Following the numerous remarks of the reviewer of the citation of Danjou et al., we scan this article for every occurrences of this citation and removed those who were unnecessary. We also tried to regroup them on a just a few necessary places and improve the references.*

*For this particular place we think that it is necessary and the removal of numerous references above makes it less annoying.*

around L210: The description would be much clearer if you would mention that the actual shape of Paris is nearly circular whereas this is not generally the case for all cities, which is why you had to adjust the radius of interest. *We removed all the parts concerning adaptation of the plume boundary definition. Indeed, it was not the plume boundary definition that changed compared to Danjou et al. 2024 but the city definition. The changes described were only the impacts of the new definition of the city boundaries, and thus not relevant.*

L211: What does "4.3" mean? Please clarify. *It is the section of the Danjou et al. in which the info can be found. Sentence has been removed following answer to previous comment.*

L215: I do not understand what you mean by "directly above the city". From my understanding, you have to estimate the emission downwind of the emission target. Please clarify. *Those are the pixels above the city. We removed this part as it was not relevant, and may partly explain the reviewers incomprehension.*

L218: As the IME abbreviation is not in the main text, it would be better if you defined it in the Appendix C where it is actually used. *parenthesis removed and addition of IME l.540 (section C1)*

L221: appendix –> Appendix C (or actually Appendix A if the Appendices A and B are removed as suggested above) *see choice made in previous comments*

L222: I think you would like to do an error reduction of your analysis, so please rephrase the title accordingly *We do not understand what the reviewer meant by this comment. However, we agree that the title needs to be expanded for clarification. Title changed to "Analysis of the sensitivities of the emission estimation error to observation conditions : general principles"*

L222: Please explain first what you mean by the "sensitvities" and what is the general purpose of this section before going into the details. *Addition at the section beginning of "To identify the main criteria of classification of the images based on the performances of the emission estimation, we analyze the sensitivity of the emission estimation error to the different variables characterising the observation conditions and the inversion. We thus can see which variables are influencing the most the emission estimation error, and define criteria, based on those variables, determining whether a pseudo-image is suitable for emission estimation or not."*

L229: You have not mentioned how the error on the emission estimate is calculated before. Please add this information to the previous section about the methodology. *A definition is now present in the introduction.*

L231: As mentioned above, it is not clear from your description why you are doing this sensitivity analysis. A motivation for this is needed. *A paragraph has been added at the beginning of the section to describe its motivation.*

L233-234: "a way to define ...": This is the motivation. Please move this to the front of the section. *See addition at the beginning of the paragraph (answer to comment about l.127)*

Sect. 4.1: I think you're saying here that you are binning your images according to percentile thresholds in 5% steps. But it is not clear from your description if you do this for each city separately. *We do not do this for each city separately, as you can see by the number of images in each bin.*

L263: "variance reduction": So your error mentioned earlier is the "variance"? Please explain. *No. This is a general description of a decision tree algorithm functionment. Here, we have removed variance reduction and instead use the term "loss function" for clarity*

L266: "the depth is set to 2": You do not mention here, which criteria are used in the end, which is confusing. It would be better to combine this discussion with Sections 5.2.1 and 5.2.2 where the criteria are discussed, also because the second choice ("diagnostic" variables) depends on the first choice. *The criteria selected by the tree are the main result of our analysis. We would rather keep the separation between the method and result section and describe the found criteria in section 5. The description of the decision tree as be rewritten and we hope that it is clearer now.*

Sect. 4.2.2: It seems as if your decision tree is a two-step procedure: First, you do it for the predictable variables, then you do it for the remaining images using the diagnostic variables. Please include this information somewhere. *Indeed it was not clear and was undermining the comprehension of this article. We hope that the developments of the introdution of section 4 answers this comment (particularly the addition of "The two types of variables are analysed separately as they can answer to two different questions. The predictable variables can be used before the inversion to determine if an image will give a reliable emission estimate and is thus worth acquiring and inverting. The diagnostic variables are accessible only after the acquirement of the image and the inversion, and can thus just give an indication on the reliability of the emission estimate.".*

L285-286: The sum of the number of variables in brackets do not match the ten diagnostic variables. *"8" replaced by "9" in the brackets.*

L286: I think it would be appropriate to put the table here as part of the main text. Otherwise, the reader has to go to the end of the paper to understand what you are referring to here. *Ok.*

L290: There are many peer-reviewed publications highlighting the importance of wind in the calculation of emissions from emission targets, so please use another publication here. *The sentence does not say that "the wind is important in the calculation of emissions from emission target", but that there is a correlation between the accuracy of the emission estimation and the mean wind speed or the wind spatial variability, which is a more precise statement. We have added a reference to Feng2016.*

L294-298: So your suggestion is to apply exactly this method with the same thresholds to real satellite data for these cities? *Yes.*

L299: You have already characterised the background $XCO_2$ in the sections before. Do you mean you want to characterise the variability of the background because this could lead to errors? *Change l. 299 and 300 by "To characterise the complexity of the background $XCO_2$ field in the image, we use the spatial variability of the $XCO_2$ concentration. This variable has been highlighted by Danjou et al. 2024 as being correlated to the error on the emission estimation. Indeed, a high variability of the background leads to an estimation of the background concentration (step (iii) of the inversion method) less accurate and thus an error in the plume enhancement estimation, and thus in the emission estimation." We hope that it clarifies the difference between this "backround variability" and the background estimation defined earlier.*

L300: It is clear that the background is crucial for the analysis of the emission plume. There are many publications highlighting this, so please use another citation for this. *We are not talking about the importance of getting a precise background estimation, but of the fact that the error on the emission estimation is correlated to the variability of the background inside an image. We have developped the description to make our point clearer. To our knowledge, not many studies pointed out this relation as clearly. We hope that the changes made following the previous comment helped clarify our point.*

L303-304: It is your choice to define a fixed size of the images. You could easily include a variability in the size of your images to analyse this effect, e.g. by adding a random parameter to the edge size of the images. Because you're saying this could be important this would be worth doing. *Indeed this would be interesting and we keep this in mind for further study. But we think that the number and type of variables studied are already important and adding one won't affect the importance of those we highlighted.*

L314: This is the very first mentioning of GP2. Please define what you mean by that. *Typo. Replaced by "our".*

L314: It is unclear from the description where this number of 92% comes from. Please extend your explanation. *Addition of ": in 8% of the cases, the optimisation does not converge"*

L315-316: Does that mean that the uncertainty of your method is 78% which would be huge and mean that we basically cannot infer much information from that? Please clarify. An illustration of this would be helpful, too. *Indeed the uncertainty is huge. Rather than saying that we cannot infer information, we would say that to infer reliable information we need a lot of images, as the method has nevertheless a small bias. Figure 3 and 5 give illustrations of the bias and spread of the method. We added l. 316 : "Reducing the bias and spread of this distribution is essential in order to obtain usable emissions estimates."*

500 L320: "the variability of the error distribution remains large across cities": Again, an illustration would be very helpful here to understand what is meant by error distribution. *The error distribution is now explained earlier, following previous comments, which should make it easier to grasp what those percentages mean.*

L324: Are you saying here that your method can only applied to single cities? *We don't understand the reviewer comment. Our method can apply to every cities in the world, but the bias and spread of the emission estimation distribution will depend of*
505 *the city characteristics and the meteorological conditions. We have modified the sentence and hope its meaning is clearer now : " The strong disparity of the error distributions between cities suggests that the error on the emission estimation is sensitive to the city characteristics (topography or city-specific atmospheric conditions) and/or to the city emissions (spatial distribution, magnitude,..)."*

L331: "significant": Did you check for statistical significance here? If not, please replace by "increased". *Replaced " are*
510 *impacted by significant errors" by "show an important bias (see Fig. 3)."*

Fig. 3 caption last line: left-hand –> right-hand *Done.*

L335: Since Appendix C has many subsections, please refer to the correct subsection here. *Reference changed to C.2.*

L337: "impairing our ability to determine the optimal set of thresholds": If the variables are correlated they do not provide additional information, so you should choose independent variables. *The point of the study is to determine which variables*
515 *are the most important between those correlated variables. We are looking for the ones that provide the most information. As we are unable to determine them using the preliminary analysis, we have decided to use a decision tree learning method. The last paragraph is replaced by "The error in estimating emissions therefore shows sensitivities, sometimes complex, to several variables, some being related, again in complex ways. Because of those intricated sensitivities, the simple analysis conducted in this subsection is insufficient to determine the optimal set of variables and thresholds for defining the most optimal*
520 *discrimination criteria for the synthetic images. This justifies the use of a more complex learning method. The supervised learning method described in Section 4.2 will enable us to determine the discrimination criteria more objectively, despite the covariances among the variables."*

Section 5.2: These results should be moved to a much earlier place, because otherwise the manuscript is hard to follow. *See our previous comments concerning the suggested reordering of the sections. We have developed the introduction of each*
525 *sections and hope that it make our study easier to follow.*

L352-357: I think what you want to say here is that you calculate the median threshold and remove all data that are beyond the threshold where the error increases. Please clarify. *Not at this stage of the study, we do not yet focus on "data that are beyond the threshold where the error increases". We made some changes to clarify. l.355 "The emission budget distribution" is changed to " The distribution of the threshold on the emission budget". "The distribution of the criteria is" changed to "The*
530 *distribution of the thresholds are". l.357 Addition of "For a given pair of criteria among the 82 retained, the subset giving the lowest error is that formed by images whose spatial variability of wind direction is below the threshold given by the decision tree and whose emission budget is above the threshold given by the decision tree. [The 82 subsets are] " l.357 Removal of ", however,"*

L354-355 and L356-357: Why is it important how many images are between the bounds? I do not see the importance of these sentences. *To see the stability of the criteria.*

Fig. 5 caption: It would be good to add the absolute emissions of the 31 cities you investigated at some point of the manuscript. Do they differ by many orders of magnitude? *addition of the emission budget on a twin x axis of figure 5.*

Fig. 5: From the description in the main text, I think the values on the y-axis are not given in percent but in ratios to the emission. *yes indeed. The label has been changed. "GP2" replaced by "gaussian plume" in legend.*

L362: Since it is not explained at all in this study up to now, where the "error" comes from, this statement cannot be validated. Please explain in detail at some point how you define and calculate the error in your analysis. *Addition of "in their framework (perfectly known background concentration, simplistic simulations of the urban plumes)". A definition on what we call the error on the emission estimation as also been added following previous comments.*

L424: Please repeat these here or somewhere else in the paper since your previous study has not been published yet. *The study has now been published in Remote Sensing of Environment.*

Fig. 6: Please add numbering to the panels and refer to them in the text. Otherwise, it is not clear which panel you are referring to. *Ok.*

L448: The orbit of OCO-3 is not really predictable so that the overpass can happen at any local time during daytime. Please clarify. *"These different times are chosen to be representative of the times when OCO-3 passes through." replaced by "These different times are chosen to sample possible times of OCO-3 overpasses".*

L449: Why are you using 11° here and not 12°? *This is an error, 12 is the right number.*

L451: It is clear that a lower resolution will result in smoothed and more homogeneous wind speeds/directions. *Addition of "and thus leads to smaller values of the spatial variability of wind direction".*

L452: As mentioned above, the resolution of your model is not 3 km but rather something around 5 km. *This was a miscomprehension due to a very poor description of the model from our side. This has now been changed.*

L453: "are located in Asia and America": Which is surprising because these cities have supposedly the highest emissions. How do you explain this? *As we say in the mentioned sentence, we are just looking at the criterion on the spatial variability of the wind direction, in this case.*

L456-460: This should be mentioned much earlier. Please mention at your earliest convenience that you are interested in clear-sky conditions only here. *Addition in the description of the synthetic images of "We do not take clouds into account when generating our synthetic images."*

L461-462: "as current instruments cannot make measurements mover both water and land in a limited time and space interval.": I think it would be better to say that the signal-to- noise-ratio is lower over water making measurements more challenging. *"as current instruments cannot make measurements over both water and land in a limited time and space interval." replace by ". Indeed, the difference in reflectivity between terrestrial and aqueous surfaces results in very heterogeneous measurement quality, and we can see for example on the OCO-3 SAMs partially overlooking aqueous surfaces that the pixels above these surfaces are often filtered."*

L463: I thought it was 20 km. Why is it 30 km here?? *We are not talking about the city limit definition. Here we are defining a "city neighbourhood" to estimate the proportion of aqueous surface in the vicinity of the city. We added "For the analysis of this subsection," to clarify our point.*

L465-468: I think this should be part of your methodology and not of your discussion of the results because this makes the connection between your model simulations and the real satellite measurements. *This analysis is much less detailed and precise and is mostly an overview to feed discussions. We thus don't think that they should be put at the same level. Furthermore, we think this will confuse the reader and make the objective less clear.*

L468: Why 11 degrees? Above, you derived a threshold of 12 degrees. *Error, corrected.*

L471: As can be seen for the cities at the west coast of middle Africa, there are regions with large cloud frequency in Africa. Please rephrase. *This sentence is clearly wrong, we are sorry for this. Replaced ", as cloud cover and spatial variability of wind direction are generally lower in Africa than in other continents" by ". Indeed, for those cities the emission budget is the discrimination criteria, and not the cloud cover nor the spatial variability of the wind direction."*

L474: "no more observable cities": Please rephrase because this means that there are no cities that can be observed. You could e.g. write something like "the number of cities does not increase..." *Done*

L476: "if observed daily": maybe better "if there were daily SAMs or overpasses" *changed to "if there were daily overpasses"*

L477: Please move this sentence above to around L466 where you describe the general procedure. *moved to l.460*

L480 and L484: "stand[s] out" is not a scientific notion. Please remove and just write the facts, e.g. "In Australia, only five cities..." *ok.*

L481: "For this contintent": Maybe better talk about the five cities instead of the contintent. *ok.*

L484-488: I think it would be much better here to talk about absolute number of cites instead of the relative number since the number varies a lot between contintents (e.g. 5 in Australia and 273 in Asia). *numbers added.*

L496: "xx vertical levels" Please include the number. In addition, the vertical resolution in the boundary layer is of relevance here. *Sorry for the typo. The number is 49.*

L497: Do you mean 137 levels? *No : https://cds.climate.copernicus.eu/cdsapp#!/dataset/reanalysis-era5-pressure-levels?tab=overview*

L500: "is less sensitive to sampling": I do not understand what you mean by that. For a given grid spacing, you get variability only for a certain resolution for all variables. Please clarify. *Typo. We replaced sampling by resolution.*

L518: "precise" What do you mean by precise? How representative are these thresholds for real applications? *We mean that we have studied they're stability to changes in the datasets and they are stable. The representativness is discuss in section 6.1 and 6.3. The conclusion has been greatly corrected and we have detailed our idea.*

L532: But one of the strengths of your method is that you can give some indication which cities can be used for satellite observations. So, which cities come out of your study whose emissions can be observed from space? This conclusion is missing here. *The cities are the ones passing the criteria. As the spatial variability of the wind direction may be seasonal and the emission budget might evolve rapidly, we think that putting cities names might distract the reader from the main point of the study.*

Code and data availability: I don't think that this is conform to AMT policy. Please add your data to a repository to be publicly available. *this will be check with the editor.*

Appendices A and B: Actually, I do not see the reason to put these figures into the appendix of the paper. They can be part of the main text. Their discussion is done in the main text only anyway at the moment. *see answers above.*

Fig. A1: Please add information to the caption how the emissions shown in the panels are calculated. *Addition of "The emissions maps are taken from ODIAC."*

Fig. A1 caption: "OLAM boundaries": Please clarify: Is this your estimated area of the city or is this the GRUMP product mentioned in Sect. 2.4? If it's the GRUMP product, I would prefer that you show the boundary of the city you use in your analysis. *This is an error. The OLAM boundaries (= Grump limits) were present in a previous version but removed. Removal of " of the OLAM boundaries of the cities and the boundaries".*

Table B1: As mentioned earlier, I'm missing the cloudiness as a parameter here which will be very important to select times suitable for satellite measurements. *See previous comments.*

L547: "in over 98%" Don't these methods vary in this number? *Yes. Addition of "for each of the three methods." As this variations is only a few percent, we do not judge it relevant to detail further.*

Fig. C1: The quantile for the emission budget is not in the range between 0 and 1. *Indeed, there was a typo in the label.*

Fig. C1: The maximum true ratio in panel (e) is 5 whereas it is 3 for the retrieved ratio in panel (d). How can you explain these differences? *As can be seen in the figure, the difference is important only for the last decile, i.e. when the anthropogenic enhancement is high compared to the background variability. We haven't explored this point, as it was not judge sufficiently relevant by the tree learning method, and as the conclusions might not be used with actual data as our modelisation of the noise is simplistic.*

Fig. C1: panel (f) please add units of the standard deviation *units added.*

Fig. C1 panel g: Where does the local minimum at the 0.5 quantile in both methods come from? *This minimum seems to disappear after filtering and is not judge as an important critera by the decision tree. The values for the optimized radius seem to be driven by two things : the spatial variability of the wind direction and the shape of the city. Given the small number of cities, we think that this pic is due to weird effect of the relationship between the two. Moreover, the statistically-speaking low number of cities does not help to have a smooth curve and increase the chance of having outliers.*

Fig. C1 caption: Please add to the caption why there are only 2 lines in panel (g).Caption of Fig. C1: "error" with respect to what? *Addition of "The optimized radius shown in panel (g) is a parameter of the gaussian plume models (see section 4.3) and is therefore not calculated for the other methods." The legend says "the error on the emission estimate" therefore we do not understand the reviewer second comment. We replaced estimate by estimation.*

L567-568: This sentence "Cities with emissions..." does not make sense to me. Please rephrase. *"important" was missing. Remove typo at the end of the sentence also.*

L571: Please remove "very" *ok.*

L572: Please add more information at which quantiles they are similar. *"is" changed to "are". Addition of "(typically above the 6th decile)"*

L573: Where does the "real" anthropogenic signal come from? *Changed to "actual" to match the previous terminology.*

L578: Please remove "very" or define its dependence e.g. by numbers. *ok*

L583-586: I don't understand these arguments. From my understanding, if the variables are correlated, only one of them should be included in this analysis in the first place. *We have detailed our purpose in section 4 following similar remarks earlier in the review.*

L590: There is no section 5.3.4.2 *reference corrected*

L590: There is no table 5.2. *reference corrected*

L592: Why is the number lower for CS and IME? *Those methods seem a bit less sensitive to the spatial variability of the wind direction. We have for example 22 occurrences of just the emission budget as criteria for the CS method.*

L593: Remove "very" here. *ok*

L598: Again, there is no table 5.2. Please update. *reference corrected*

L608 and L609: I do not understand what you mean by "standing out". Please rephrase and explain. *"without standing out strongly (appears for less than half of the samples)." replaced by " As this criterion appears for less than half of the samples, we do not consider it as sufficiently relevant."*

L611: Why do you now use the same thresholds as for GP2? The thresholds will depend on the method you use. In the previous section, you showed that the thresholds are different for each method and to use the optimal one for each method would be the way to go. *We do this to simplify. The differences are furthermore quite thin : they all fall in the spread of the thresholds found for GP2. We thus think that it should tend to converge with a bigger dataset, more various data. An explanation has been added : "As the thresholds distributions are similar for all inversion methods, we choose to use the same threshold values than those found for the GP2 method".*

L611: The description in section 5.2.1 has never been referred to as GP2 method. Please add this to this section. *Addition l.545 "The gaussian plume method used in the main body of the article will be now referred as GP2 for clarity."*

Fig. C2: Please add the unit to the y-axis *Done.*

Fig. C2 caption: Do you mean emissions > 2.1 ktCO$_2$/h? *Yes. Correction made.*

L614: What do you mean by "accuracy"? Is it the median or the spread or both? *Spread. Changed to "spread of the error generally decreases with higher cities emissions".*

**1.3  Technical corrections:**

L28: Please use citep instead of citet *ok*

L29, L37, L38: XCO$_2$: subscript for number 2 *ok*

L30: citep instead of citet *ok*

L59: remove second "the" *ok*

L61: "is" –> are *"errors" changes to "error"*

L83: help to identify *ok*

L84: help to identify *ok*

L88: Remove "The" at the beginning of the line. *ok*

L88: move "used" before OLAM simulations *ok*

L97: Use (**?**)or Ullrich citation *ok*

L116: I think, it should be "power plants" *ok*

L145: a –> an *ok*

L154: 2063 km$^2$ (squared is missing) *ok*

L156: Remove comma between "Note that" and "the" *ok*

L161: Use **?**or the citation *citation removed*

L163: citation is wrong *ok*

L201: It would be better to include commas between the vector elements. *ok*

L214: annex –> Appendix *ok*

Figs. 2 and 3: Please switch the order of the figures to match the mentioning in the text. *Early reference to fig 3 was removed. The order is now correct.*

Sect. 4.2.1: Please convert the verbs from future to present, e.g. "will separate" –> separates *ok*

Fig. 2 caption: will classified –> will be classified *ok*

L294: "are based" –> is based *ok*

Fig. 4: black line –> black dotted line *ok*

Fig. 4 caption: the the –> the *ok*

L397: modelisation –> simulation *we do not agree with this correction*

Fig. 5: typo in y-axis caption: % of the true emissions *ok*

Fig. 6: right panel x-axis: Better "meeting" instead of "combining". *ok*

L476: "an order of magnitude of" –> approximately *ok*

L484: "of cities of" –> of cities with *ok*

L504: "5.2.2" –> "Sect. 5.2.2" *ok*

L511: "XCO$_2$" add subscript *ok*

Fig. A1: Please add more space for the Ningbo panel. *ok*

Fig. A1: Please reverse the color scale because it is confusing that red means lowest emissions. *The plots seemed to us less lisible with the inversed colormap suggested by the reviewer. We prefered to keep it that way.*

Caption of Fig. A1: Add "Sect." to 2.3 *ok*

Caption of Table B1: 1first –> first *ok*

Caption Table B1: W2D –> W (which actually occurs in the table) *No, we are talking here about the wind use for the estimation of the divergence, shearness,.. We agree that the notation was confusing and changed it.*

L538: "optimized" –> "applied" or "investigated" *"investigated"*

L539: "across" –> for *replaced by "of"*

Caption of Fig. C1: remove "according" *ok*

L568: Remove "emissions." at the end *ok*

L598: close –> similar *ok*

---

## Author Comment (AC2)

**Review of "Optimal selection of satellite $XCO_2$ images over cities for urban $CO_2$ emission monitoring"**

Alexandre Danjou[1], Grégoire Broquet[1], Andrew Schuh[2], François-Marie Bréon[1], and Thomas Lauvaux[1,3]

[1]Laboratoire des Sciences du Climat et de l'Environnement (LSCE), IPSL, CEA-CNRS-UVSQ, 91191 Gif sur Yvette, France
[2]Cooperative Institute for Research in the Atmosphere (CIRA), Colorado State University, Fort Collins, USA
[3]Molecular and Atmospheric Spectrometry Group (GSMA) – UMR 7331, University of Reims Champagne Ardenne, 51687 Reims, France

**Correspondence:** Alexandre Danjou (alexandre.danjou@lsce.ipsl.fr)

*We would like to thank both reviewers and the editor for the careful and detailed reviews and their patience during this process.*

*It seems to us while reading the reviews that the aim of the study and of the different analysis conducted were not sufficiently explained, which perturbed the reviewers and undermined the comprehension of the links between the different parts. We thus have made numerous changes on the introduction and took time to better introduce each sections. We have also changed the title as we found that it might give the impression that we used a model for our inversions, whereas we only use our model to generate our synthetic images. The description of the learning method has also been completely rewritten to make it clearer.*

*Multiple comments were in relation to the article Danjou et al. 2024. This article is now published and accessible (https: //doi.org/10.1016/j.rse.2023.113900). Some of the references to it were unecessary and have been removed. We also tried as much as possible to gather the references and explain them when necessary. We hope that the article is now more self sufficient in regard to Danjou et al 2024.*

**1    Reviewer 2**

The manuscript addresses an important scientific issue and presents an innovative approach to assess CO2 emissions from urban areas. However, many parts in the preprint suggest that it is work in progress. The study holds potential but necessitates substantial revisions to strengthen its clarity as well as scientific rigor.

*We would like to thank the reviewer for this complimentary comment. We hope that the corrections made will address his concerns. We have rewritten the introduction and conclusion and added introduction parts to the section and hope that it had helped clarify the article.*

**1.1    General comments:**

The manuscript should make a clear distinction that its primary objective is not the inversion process itself, but rather the assessment of the confidence in the inversion results via predictable and diagnostic variables. In the methodology section,

more emphasis could be put on how the two-step procedure via predictable and diagnostic variables contributes to the accuracy assessment of emission estimates. *Indeed, this was missing. We have developed the introduction of this section to make this point clearer.*

25   The manuscript heavily relies on Danjou et al. 2024, which is not accessible for verification. I think this is problematic and impedes a thorough evaluation in parts of this study.

*It seems that there as been a problem. The manuscprit of Danjou et al. (2024) was provided to the editor so that it can be send to the reviewers. Please accept our apologize for this. The article has been published in RSE and is now accessible at https://doi.org/10.1016/j.rse.2023.113900.*

30   The manuscript lacks clarity on what constitutes the error. The reference data used for determining the error is not explicitly defined. *A definition of the error as been added l.76 : "As we are working with synthetic data, the error in the emissions estimate is directly accessible by comparing the emissions estimated by the inversion method with the synthetic true emissions used in the OLAM simulations."*

The manuscript does not provide validation of its results with actual satellite data (e.g., OCO-3 SAM) or ground truth
35   measurements. The absence of this validation makes the reliability of the simulated results vague to some extent.

*The use of synthetic data is often used alone (Broquet 2018, Lespinas 2020, Kuhlmann 2019) to validate methods as it gives a good idea of the expected behaviour of these methods. There are indeed always differences between real and synthetic data, and differences in sensitivity will certainly be visible with real data, as there is still much to be understood in $XCO_2$ measurements at fine scale. Moreover, correlation unreprensented here can also appear that will highlight other criteria. Nevertheless, the*
40   *criteria found here will still be pertinent with real data, are they will still be driving the error. From our point of view, comparing real and synthetic data (and their sensitivities) merits a study in its own right.*

The variability of the error distribution remains significant across different cities. What are the implications for estimations from real satellite images? *It means, as stated in the conclusion, that there are still things that we do not understand, despite the progress made by defining these criteria. We have developed the conclusions to answer this question.*

45   The study should discuss the detection limits of current and future satellite missions and how those might impact the results. Is the purely random noise model imposed on XCO2 data in the study representative of real world atmospheric and environmental conditions? In actual scenarios, factors like surface reflectivity of different land types and the presence of aerosols can introduce more structured or systematic errors rather than purely random ones. Would incorporating more realistic, structured errors enhance the model's applicability and accuracy in real-world urban CO2 monitoring scenarios? Given this potential
50   limitation, what implications does this assumption have for future research?

*Understanding and simulating the error structure is an active field of research (see Bell 2023 https://amt.copernicus.org/articles/16/109/2023/, Taylor 2023 https://amt.copernicus.org/articles/16/3173/2023/amt-16-3173-2023.pdf , Worden 2017 https://amt.copernicus.org/articles/10/2759/2017/). We think that the complexity necessary to accurately take this into account (as can be seen in the papers cited) is out of scope of this study. Integrating this would only add complexity, increase the*
55   *error and maybe add some new criteria, but the criteria that we find in this study should still be pertinent.*

The selection criteria for the size of the target emission zone radius, are not comprehensively described. I can't find a clear rationale for the chosen size of the emission zone radius. Any potential to estimate this radius through an inversion approach? *Indeed, a clear explanation was missing. We have reshaped the first paragraph of this subsection and introduced a rationale.*

The influence of cloud coverage on satellite observations is a significant factor that the manuscript should address. *When it comes to determining which cities are most suitable for measurement, cloud cover is indeed a major issue, and the frequency of cloud cover is an important criterion for city selection. We examined this point in section 6.2.*

*We have reserved this analysis for discussion, however, because realistically adding the effect of clouds in images (filtered pixels and contamination of neighbouring pixels) and trying to objectively quantify their impact on error is of a complexity that would merit a dedicated study to do it more properly than what is proposed in the discussion.*

*We also believe that, once an image is partly contaminated by clouds, it is not worth processing. Indeed, the effect of cloud presence on the measurement is very complex, due to the presence of 3D cloud radiative effects in OCO-2 retrievals (https://doi.org/10.5194/amt-14-1475-2021). Given the size of these effects (of the order of a few kilometers) and the size of our images (a few tens of kilometers), we doubt the value of trying to obtain an estimate from an image contaminated by clouds. However, this remains to be demonstrated, and the complexity of such a task requires, in our opinion, a complete study and has no place here.*

*Adding clouds would not alter the conclusions we have reached, the criteria identified should still be important even in the presence of clouds. Thus, adding cloud coverage would only increase the error and add another criterion.*

The authors of the manuscript should revise Section 3 to summarize only the key aspects of the Danjou et al. 2024 study that are directly relevant to their current research. As mentioned above, it is problematic that the Danjou et al. 2024 paper is not yet available and that the study heavily relies on it. *Danjou et al. 2024 is now published. We tried to remove the unecessary references to this paper and regroup the ones that were unevitable. We hope that this article is now easier to read and less repetitive from that point of view.*

The description of the OLAM model and its simulations should be more comprehensive in order to better understand the simulation results. *Changes have been made following the specific comments of reviewer 1.*

The manuscript acknowledges that the method is not universally applicable to all cities and is still a work in progress. However, it should also describe how the proposed method would be applied to real satellite measurements and critically assess, whether its applicable at all within the error budget. *We do not share the conclusion of the reviewer. Could the reviewer indicate which sentences give this impression? Indeed a gaussian plume is applicable to all images and the criteria are also valid for every cities.*

The conclusion should reiterate the study's focus on developing a procedure for accuracy estimation, summarizing how the study advances this goal and its implications for future research and practical applications. *We have rewritten the conclusion, keeping in mind the reviewer advices.*

**1.2 Specific comments**

The manuscript frequently uses abbreviations and technical terms without defining them (e.g., XCO2, UNFCC, OCO, ppm, GOSAT-GW, WRF, OLAM, IQR). *definitions added in the relevant places.*

3: The phrase "selecting images to be processed" should be more clearly defined. *changed to "selecting the images to be processed"*

Fig 1: Identifying the factors behind the peak XCO2 values in the simulation domain deyond the city boundaries. *The two small peaks on the east of the city may correspond to an XCO2 enhancement due to the city that as accumulated earlier in the day and is now ventilated. The peak on the bottom of the images seems to be coming from the city at the edge of the image when we look at a more broader view.*

46: The Danjou et al. (2024) should be made available to the reviewers due to its significant relevance to the research. *It should have been. Sorry for this issue.*

53: Clarify local background signal (not clear to me, what is meant by this term). *it is the background signal in the vicinity of the city (please see Schuh 2021 for a more accurate description).*

56, 58: "This study" can be misleading. Write "The study". *Changed to "Their study"*

48-50: Consider moving the sentence up to line 25, before you start describing the OCOs. *The mentionned sentences have been removed as they weren't bring new informations compared to the ones at l.25.*

261: Set by the user? Do all produce the same value? *The description of the decision tree as be rewritten and we hope that it is clearer now:*

*"4.2 Analysis with the decision tree learning algorithm*

*In this study, we seek to better understand the relationship between the input variables (predictable/diagnostic variables) and the reliability of an emissions estimate. For this, we train an explainable machine learning algorithm to predict the relative error of the emission estimate given some input variables (described in Section 4.3), like the variability of the wind direction or the emissions budget, and then study which variables are determined to be relevant by the algorithm. We choose a regression decision tree for this, as they work by learning simple decision rules and therefore are highly interpretable while able to find non-linear relationships between the inputs and the target variable.*

*4.2.1 Description of the decision tree learning algorithm*

*A decision tree is constructed following a recursive process: at each step, the algorithm splits the data into two subsets following a binary rule on a single variable, finding the split that best reduces a particular loss function on the target variable. Each subset is split further into two until some stopping condition is reached (see Fig. 2 for illustration). This algorithm therefore splits the input space into regions, where each region corresponds to a similar value of the target variable (i.e. the error on the emission estimation in our case). We use the regression tree implementation from the scikit-learn library (Pedregosa et al., 2011) with a squared error loss, and impose conditions on the algorithm to prevent overfitting (creating over-complex trees that don't generalise well): we set the maximum depth of the tree to 2 (i.e. two levels of binary splits) and*

*we impose that the leaves must contain at least 10% of the training set. The training set (at the root node) is described in the following paragraph."*

405-406: Please clarify why Paris exhibits such a low emission rate? Is this due to the absence of significant point sources? *Indeed there are no significant point sources in Paris. However we do not think that discussing the low emissions of Paris is relevant in this study, are we are working with synthetic data and are not interested in city in particular.*

418: What are the implications or consequences for the study if the dependencies of errors are not completely comprehended, even when using synthetic data. *This study brings objective criteria to evaluate the accuracy of an emission estimation, criteria that have not been evaluated as precisely until now to our knowledge. The error distribution of the emission estimation has not yet been (and cannot be?) fully characterized. Thus we will still have a non negligeable range of uncertainty on the bias given by an emission estimation for a certain city. However, the criteria found here remain pertinent and will still be pertinent when applied to real data. We have adapted the conclusion to emphasise this.*

Fig. 5: Typo in y-label (thrue). *ok*

316: The GP2 inversion method is presumed to be a variation of the Gauss Plume approach. However, at this point it is unclear for me what the '2' in 'GP2' represents. Please provide further details or clarification. *It is a typo remaining from an older version of the draft. There was a GP1 (a variation) that we discarded. We thought we had removed this naming in the main body. It is now done.*

520: Was the 12° threshold for wind variability found empirically or is there a rationale behind choosing it? Is this higher variability threshold, compared to what's mentioned in Danjou et al. 2024, a result of the increased resolution in the model? *The threshold of 12° is the one found in section 5.2 of this article with the tree learning method. We have rewritten this paragraph of the conclusion and , among other things, replaced l.518 "We were also able to determine precise and objective thresholds on these criteria to select the images." by "This analysis with a learning method also provides precise and objective thresholds to these criteria supporting the selection of images."*

*The threshold that the reviewer refers to may be the 7° threshold defined in Danjou et al. 2024. It was empirically defined, which explains the difference. Addition of "(empirically defined)" l.317*

578: The error appears to be highly sensitive to the city's radius. Could you clarify what "pseudo-image filtering" specifically entails? I don't have access to the Danjou et al. 2024 source for reference. Additionally, referring to line 550 and following, I guess it implies filtering out scenes with variability above a certain threshold? *This minimum seems to disappear after filtering and is not judge as an important critera by the decision tree. The values for the optimized radius seem to be driven by two things : the spatial variability of the wind direction and the shape of the city. Given the small number of cities, we think that this pic is due to a weird effect of the relationship between the two. Moreover, the statistically-speaking low number of cities does not help to have a smooth curve and increase the chance of having outliers. We rephrase l.551 and l. 578 to clarify and make the statements more self sufficient in regard to Danjou et al. 2024.*

58: What are pseudo-images? Synthetic 2D CO2 concentration images, I guess? *yes. Precise definition added l.51 : "synthetic satellite images (i.e. pseudo-images) of $XCO_2$ concentration generated with a meteorological model". We have replaced every occurences of "pseudo-images" with "synthetic images" for clarity.*

229: What is the primary source of error in the emission estimates? Does it originate from the tree model or the GP2 inversion method? *The tree model is not used to estimate the emission but to classify the emission estimates. The main sources of error are the estimations of the background estimation and the wind speed (cf Danjou et al. 2024).*

47: Improve the English language in this sentence. It encapsulates the primary motivation of the study and thus should be distinctly emphasized and articulated. *We do not understand what is wrong with this sentence as it seems to us grammatically correct.*

76: The frequent citation of Danjou et al. 2024 for all the "light" methods seems inappropriate. It would be more suitable to refer to the original papers specific to each approach. *It seems that the line given by the reviewer is not the good one. On line 74, we only cite Danjou et al. 2024 as we say that we are using the same methods as therein. We assume that the reviewer is refering to the paragraph between l.36 and 51 or the annex. We removed the unnecessary references to Danjou et al. 2024 in this paragraph and we added the relevant citations in appendix c1.*

74-79: The logic presented in this paragraph is unclear. It would be beneficial to revise and clarify the content for better understanding. *We have completely reshapen and rewritten the paragraph, with some additions to make it clearer : "The objective of our study is to resume the series of analysis from Wang et al. (2018); Schuh et al. (2021); Danjou et al. (2024) and deepen the evaluation of the conditions corresponding to reliable estimates of urban $CO_2$ emissions using satellite $XCO_2$ images. To do this, we use a little more than a month of simulations of local $XCO_2$ scenes over large cities. This simulations are generated with the global OLAM model and evaluated by Schuh et al. (2021). We use these simulations to generate synthetic satellite images for the selected cities, and estimate their emissions by applying one of the automated and computationally-light inversion methods implemented, tested and optimized by Danjou et al. (2024). By using realistic simulations to derive the synthetic image and using a method independent of the model used for the simulations to estimate the emissions, we take into account realistically the uncertainty in the meteorology, atmospheric transport and background. As we are working with synthetic data, the error in the emissions estimate is directly accessible by comparing the emissions estimated by the inversion method with the synthetic true emissions used in the OLAM simulations. The study of the emission estimation error for different cities and weather conditions aim to support the identification of criteria for discriminating between images, separating those whose processing yields statistically reliable estimates from those whose processing is statistically unreliable."*

Sec. 2: Mention that ECMWF ERA-5 product is used for meteorological data. It is only mentioned in line 446. *ERA-5 is only used for the analysis in the subsection 6.2, not before (and therefore not in section 2). l 445 "For this analysis" changed to "For the analysis conducted in this subsection".*

130: What is the methodological rationale behind assuming constant emission rates? Do you expect that incorporating dynamic emission rates would significantly alter the study's results or conclusions? *We did it for the sake of simplicity. We don't think it will altar the results and conclusions, as mentionned in the discussion (l.425-437)*

217-221: Does the observation that all methods yield similar results suggest that the assumptions in the model used to generate the synthetic data might be overly simplistic? *No. It was already the case in Danjou et al. 2024. It is explained by the fact that all method share most of their steps : same way of estimating the plume boundaries, the background concentration.*

190 *The final step (using a gaussian plume or a mass balance method to estimate the emission) has a minor impact on the overall error (as shown in Danjou et al. 2024).*

Sec. 3: Does this paragraph solely describe the work done in Danjou et al. 2024? Certain statements, like in line 160 "The method used here...", are ambiguous in the current context. Does "here" refer to this study or to Danjou et al. 2024? It may be beneficial to condense the paragraph preceding Eq. (1) for clarity. *Addition at the beginning of section 3 of "The complete*
195 *description of the inversion method and the details and justifications for its specific configuration and implementation can be found in Danjou et al. (2024). We make the assumption that the configurations chosen in the framework of their study remain optimal for other cities. This assumption seems justified, as the chosen methods for each steps differ from the discarded methods on objective criteria. This section only gives an overview of the different steps and the adaptations (compared to the reference configuration from Danjou et al. (2024)) that were made in the context of this study." We didn't shorten the part before equation*
200 *(1) as we think that the informations given here are necessary to have an idea of the inversion method used and understand the changes made compared to Danjou et al. 2024.*

217-221: Consider clarifying this point earlier in the document, perhaps where the Gaussian plume inversion (Eq. (1)) is introduced, for better coherence and understanding. *Indeed. It comes much too late. We rewrote the introduction of the article and of the subsection to make this point clear from the start.*

---

## Author Response (AR2)

**Review of "Optimal selection of satellite $XCO_2$ images over cities for urban $CO_2$ emission monitoring"**

Alexandre Danjou[1], Grégoire Broquet[1], Andrew Schuh[2], François-Marie Bréon[1], and Thomas Lauvaux[1,3]

[1]Laboratoire des Sciences du Climat et de l'Environnement (LSCE), IPSL, CEA-CNRS-UVSQ, 91191 Gif sur Yvette, France
[2]Cooperative Institute for Research in the Atmosphere (CIRA), Colorado State University, Fort Collins, USA
[3]Molecular and Atmospheric Spectrometry Group (GSMA) – UMR 7331, University of Reims Champagne Ardenne, 51687 Reims, France

**Correspondence:** Alexandre Danjou (alexandre.danjou@lsce.ipsl.fr)

I would like to thank the authors for their revisions, which improved the descriptions and the understandability of the study. However, some of the authors' responses to the reviews need more explanation and discussion in the manuscript which is why I still recommend major revisions before publication. The comments can be found below. The line numbers refer to that of the revised manuscript.

*Thanks for those encouragement and compliments. We realize that we might have misunderstood some of the reviewer comments in the first review and thanks the reviewer for taking the time to come back on them.*

Re their responses on the reviewers' comments

The authors asked in their responses where in the manuscript it is written that the method is not universally applicable to all cities and is still work in progress. The answer to this is e.g. in line 344: "However, despite the application of this criterion, the variability of the error distribution remains large across cities." and in the conclusions line 559: "These significant remaining biases raise the question of the current reliability of the results obtained on a single given city." So the comment by Reviewer 2 is still open: "However, [the manuscript] should also describe how the proposed method would be applied to real satellite measurements and critically assess, whether it's applicable at all within the error budget."

*Thanks for the clarification. We didn't understood the first time what meant the reviewer but do now : indeed the methods are*

*not sufficiently reliable to be applied blindly to actual data as things are. We add some sentences in the conclusion concerning the further work that we think necessary to develop this method and thus, we hope, answering the reviewer comment : "Future work should focus on determining the types of information that can be reliably derived considering the current error estimates (e.g. annual emissions budget, trend detection, ...) along with the required number of images/plumes following Kuhlmann et al. (2019). In parallel, applying this sensitivity analysis to actual satellite data, similar to the synthetic images used in our study*

*(e.g. OCO-3 SAMs), would help to evaluate and to refine the criteria derived here."*

Re response by the authors about the interpolation methods used: There is no "classic" interpolation as mentioned by the authors in their response to the reviewers' comments. So which method is used? Please add this information somewhere around line 140.

*The information has been added by replacing the following sentence "The results of these simulations are then projected*
*onto a regular grid at approximately* $1\text{km} \times 1\text{km}$ *resolution. This is done to simplify the analysis of the model outputs." by the sentence "The fields are then horizontally regridded to* $1\text{km} \times 1\text{km}$ *using rasterization techniques where the center of the 1km by 1km grid cell is mapped onto the hexahedral grid average which contains it. While the mapping isn't strictly mass conserving, the errors should be relatively small and since this is a post-hoc operation, errors do not accumulate during the simulation."*

Around Fig. 4 and its results: The threshold of 2.1 ktCO2/h is only based on the city emissions from the inventory. But there is no discussion whether the current satellites are able to measure emissions in this order of magnitude. Since the manuscript is based on OCO-3 and CO2M (e.g. random noise according to the precision requirement of 0.7 ppm of CO2M) there has to be a discussion somewhere in the manuscript which is the lower limit of emissions that is observable by the satellite. Otherwise, this threshold is fine in theory but not at all applicable in practice. For instance, typical parameters could be used to convert the 35 emission of 2.1ktCO2/h to XCO2 enhancement using your equations as done in the literature, e.g. [1], [2]. If this is larger than the noise of the instruments ( 0.7 ppm), then it is fine, if not, the emission corresponding to a 0.7 ppm enhancement has to be used as the emission threshold in the study.

[1] https://amt.copernicus.org/articles/4/1735/2011/amt-4-1735-2011.pdf

[2] https://www.sciencedirect.com/science/article/pii/S0959652623006832

*We still think that the notion of detection limit is irrelevant with our methods. But we think that this second mention of the reviewer to this shows some unclearness from our side. However, lacking any distance, we are unable to identify which points lack clarity. We thus try to expand our answer hoping that the explanation will make our point clearer and help us, with the help of the reviewer, identify the point that were missed. This (long) answer focus first on how we took into account instrumental noise in the threshold determination (1) , then on the questionable notion, in our particular case, of "detection limit" (2), before* 45 *presenting the analysis asked by the reviewer (3). We would be interested to hear more from the reviewer on this issue, as it is a major point in our analysis.*

*(1) The threshold of 2.1 ktCO2/h is determined by the decision tree, which is based on the results of analysis of our synthetic images. These synthetic images include noise (our image is the sum of three "signals": the city plume, background concentrations -other anthropogenic sources, biogenic sources, mesoscale variations- and 0.7ppm noise) so this criterion is determined* 50 *by taking instrumental noise into account. This value is thus a lower limit of emissions that are quantifiable with "acceptable" precision by the satellite.*

*(2) We would like to remind the reviewer that the issue when we target cities is not plume detection, but the capacity to perform emission quantification. Indeed, we do not need to detect the plume : we know where the city is and we know the wind direction, which means that we can know the position of the plume without looking at the image as shown in Danjou et al.* 55 *(2024)'s sections on plume detection methods. Those sections shows that emission estimations done with an a priori defined plume are way better than those done with a plume detection algorithm. Thus, a close and more relevant notion would be "quantification limit" : limits under which we cannot properly quantify the emissions. Our analysis tends to show that a more*

*relevant criterion than emission budget for quantification limit is wind stability in the PBL (with a criterion on the spatial variability of wind direction, or wind speed), cf last paragraph of section 5.2.1.*

*(3) The analysis proposed by the reviewer with the Gaussian model is, from our point of view, a simplified version of what we have done here :*

      – *"typical parameters could be used to convert the emission of 2.1ktCO2/h to XCO2 enhancement using your equations as done in the literature, e.g. [1], [2]" : rather than using a Gaussian model, we do this with a transport model with complex meteorology, and for several emission values;*

– *"If this is larger than the noise of the instruments ( 0.7 ppm), then it is fine, if not, the emission corresponding to a 0.7 ppm enhancement has to be used as the emission threshold in the study." : rather than starting out with an a priori view of the result (the main limitation of quantification methods is instrumental noise, and we need to find a law to determine a threshold emission limit), we use a learning method to determine these criteria in a broader context by including several parameters in our analysis.*

[Figure]

**Figure 1.** Maximum (first line), quantile 80% (middle line), median (third line) enhancement simulated by the Gaussian plume in the analysis area (see section 3) as a function of wind speed ($W$) and emissions ($Q$) for different Pasquill parameter values. The red line delimits the area where enhancement is greater than 0.7ppm.

*Using the equation given as an example by the reviewer (used in the 2 articles but also by our inversion model) also shows that an enhancement bigger than 0.7ppm will be the result of a law involving emissions and wind speed ($XCO_2 \propto Q/|W|$,*

*see Section 3 for notation) and not just a threshold on sole emissions. An illustration of the law linking ppm increase and wind speed/emission according to the Gaussian model is shown in figure 1.*

*We can confirm this first-order proportionality between the enhancement and the emission/wind speed ratio. The definition of detectable enhancement is, however, not obvious: should we base it on the median XCO2 signal in the plume, the maximum, or a specific quantile? We can see that the number of accepted cases would greatly vary from one metric to another.*

[Figure]

**Figure 2.** Distribution of synthetic images according to the average wind in the PBL at the time of the image and the city's emissions. A differentiation is made between cases accepted (blue) according to the criteria defined by the decision tree and those rejected (orange). The grey lines show demarcations at 0.7ppm for different metrics characterizing the enhancement and a Pasquill parameter of 213. These demarcations and axes are the same as those shown in figure 1.

    *Figure 2 shows the distribution of the 9,119 cases as a function of wind speed and emissions, with a differentiation between those accepted and those rejected according to our decision tree method. We can see that the decision tree criteria have no connection with the 0.7ppm lines of the previous graph (in grey on this figure).*

*Figure 3 is much the same as figure 2, except that the x-axis is now in logarithmic scale (for greater visibility) and the color of the dots now indicates the error on the emission estimate (in %). We can see that cases whose enhancement is greater than 0.7ppm according to the Gaussian plume model present errors of more than 60% (red ellipse). Conversely, cases with enhancement of less than 0.7ppm according to the Gaussian plume model show errors of less than 40% (majority of cases in the blue ellipses). These zones correspond roughly to those given by the criteria of our decision tree method.*

*The aim of this article is precisely to delve a little deeper and go beyond the general idea of an urban plume detection threshold. For us, this comment shows that our approach may not have been fully understood, and therefore requires further*

[Figure]

**Figure 3.** Emission estimation error for the 9,119 synthetic images in function of the average wind in the PBL at the time of the image and in function of the city's emissions. The grey lines show one example of demarcation at 0.7ppm from figure 1 and 2.

*explanation. We thus have added a paragraph on the notion of "detection limit" at the end of section 6.3 and a sentence in the ante-penultimate paragraph of the introduction .*

Sect. 5.1: Since this analysis motivates the use of the machine learning algorithm described in Sect. 4.2, I still think that it would be better to move this between Sect. 4.1 and 4.2. In addition, the reader has to always move back and forth between 4.1 and 5.1 to understand all the details mentioned in both sections. I understand the authors' argument that it is common practice to describe methodology first, but I think it is different when the methodology depends on previous results. And also from the point of view of the reader it would make the manuscript better understandable.

*We still disagree with the reviewer on the solution to this issue. A third of section 4 (l.234-246, l.304-338) is common to both analysis. Separating this would require a huge amount of reshaping : move 5.1 between 4.1 and 4.2 as suggested by the reviewer, but also move 4.3, and rewrite introduction parts of sections 4 and 5 (half a page each).*

*Mostly, we fear that the development it would need to make the preliminary analysis stand alone would have to be important and make the preliminary analysis take a lot of place compared to its interest. For now the decision tree analysis occupies 6 pages, the preliminary one 2 pages and the common parts 2 pages. Reshaping would need to move the common parts in the*

*preliminary analysis and thus have 4 pages vs 6. Moreover, we think that then it will also be harder for the reader to make the separation between what is common to both analysis and what is different, making it thus less clear than it is now.*

*We spent a lot of thinking on this both while writing the article and during the review. We agree with the reviewer that this choice of ordering is not perfect and make some points hard to understand, but we don't have any better solution. If the reviewer still disagree, we will make the changes in order to move on.*

*Another solution might simply be to remove the preliminary analysis which was merely here to introduce the concept of our analysis and make the decision tree purpose easier to understand. The preliminary analysis could then be put in annex under the title "Illustration of the sensibilities of the error on the emission estimation to different variables characterising the observation conditions and the inversion". But we do think that this preliminary analysis helps the understanding of the decision tree analysis and removing it from the main text would also hampers the clarity of the paper.*

Fig. A1: I still think that because the description of this figure is in the main text, it should appear there (Sect. 2.3 or 2.4). Or at least there should be a text describing the results in Appendix A. Currently, it is just the figure without any description.

*See answer to specific comment.*

**1 Specific Comments**

- L13: In the main text it is always 2.1, not 12.1 ktCO2/h. *Typo corrected.*

- L79: Please add "about" before 2km x 2 km because it varies with the distance to nadir and the ratio will not be exactly a square. *Correction made.*

- L88: It is not clear what "realistic" means here. Is realistic in the sense of the OLAM model applied? Is it the grid spacing which is similar to the expected satellite footprint size? *Both. We added a parenthesis to precise our point : "(as obtained from a global non-hydrostatic atmospheric model with a maximum resolution of a few km)"*

- L138: PBL not defined. *We replaced "PBL" by " Planet Boundary Layer (PBL)" on l. 138.*

- Sect. 2: There has to be a note somewhere that your 40-day simulation is free-running because then the forecast skill after 14 days is small and the meteorology is not realistic (but of course consistent within the model). *We added a sentence ("The simulations are free running.") in section 2.2.*

- L149: Why are the dates different from the previous manuscript? Are the authors sure that the simulation did not start
at 01 August? That would also be consistent with 8 images per day for 31 cities. *this is a typo, the correct begin date is indeed 1 August.*

- L154: "that expected for CO2M": This is only true if the city is in the center of the nadir swath of CO2M. Please clarify and rephrase. *Following the reviewer comment, we add extended the sentence to clarify : "This size is halfway between that of the OCO-3 images and the expected swath of CO2M in nadir mode."*

– L169: "most emitting pixels": As pixels cannot emit themselves I think the authors mean the pixels with largest emission sources. Please clarify and rephrase. *The sentence was rephrased into : "Within this $50\mathrm{km}$-radius disc, we select only a fraction $(1/2.5^2)$ of the pixels, keeping those for which the emissions are the highest."*

– L286: "We thus obtain at most 4 subsets": It's not really clear where the number 4 is coming from. Is it because the decision tree has a depth of two and the whole set of 9119 images is split into two subsets in each step? A reference to Fig. 2 would be helpful here, I guess. *We change the sentence following the reviewer suggestions :"As the maximum depth of the tree is two, we obtain at most 4 subsets (see illustration of that case on Fig. 2) and select the one with the smallest Mean Absolute Error (MAE) on the emission estimate."*

– Fig. 2: "pseudo image" should be replaced by "synthetic image" here as well for consistency. *Correction made.*

– L315-317: This citation looks weird although I realize that this the recommended citation. Maybe this can be abbreviated in some way? *As pointed out by the reviewer, it is the recommnded citation. We didn't find any other ways of citing it in the literature.*

– L338: "the optimisation does not converge": Do the authors mean the minimization process described in Sect. 3? What is the criterion for convergence? *We replaced this bit by : "the optimizer used for the minimization described in Section 3 does not converge." The criterion for convergence is arbitrarily fixed. We set a tolerance of $10^{-5}$ times the*

*root-mean-square-difference between the mass per unit area simulated by the Gaussian model (with the prior guess) and the "observed" mass per unit area. According to the optimizer documentation : "The iteration will stop when $\max|\mathrm{proj}g_i|i=1,...,n <= \mathrm{gtol}$ where $\mathrm{proj}g_i$ is the i-th component of the projected gradient" and $\mathrm{gtol}$ our arbitrarily fixed tolerance.*

– L369: What is GP2? *This is a remnant of the first review. Change to "our inversion method".*

– Fig. 4: replace pseudo-images by synthetic images *Correction made.*

– L443: It is not clear what is meant by "accuracy" here. Is it the bias or the spread of the distributions shown in Fig. 5? Please clarify in the manuscript. *Indeed, the term was not clear, we changed the sentence to : "We can see (figure 5) that the spread of the error on the emission estimation generally increases with decreasing emissions budgets."*

– Fig. 6: panel (a) still saying 11 degrees as the threshold. *Correction made.*

– L515: I think these percentages are the fraction of cities that pass all the criterions. I don't agree with the statement that their emissions are "easier to quantify than cities on other continents" because this is not relative to other continents. Please rephrase. *Indeed, it was misleading. We changed the sentence to "Asia and Australia stand out, with 37% (102 cities) and 40% (2 cities) of cities passing the criteria. Indeed, those cities, according to ODIAC dataset, are more likely to have emissions above our threshold."*

– L518: 0.7 ppm is the precision requirement for CO2M, not accuracy of OCO-3. Please clarify and rephrase or provide references here. *We changed the sentence to make it refer to CO2M. In the previous version, we had the article of Worden et al. (2017) in mind, which demonstrated a precision and accuracy of around 0.7 ppm for OCO-2 (which is equipped with the same instrument as OCO-3). But we agree that this is not indeed directly applicable to OCO-3 and changed the reference.*

– L528: This sentence still suggests that ERA-5 has only 37 vertical levels, which is not true. The authors should add here that they are using the ERA-5 product on constant pressure levels. By the way, it would be better to use the native resolution of the reanalysis with 137 vertical levels to ensure that the best resolution within the boundary layer is used, which is most relevant for this study. *The sentence was changed to clarify which product we are referring to. We also clarified that our main point is about the horizontal resolution of the data : "Indeed, the horizontal resolution of the*
*weather product used here is very high around the cities of interest ($\approx 3\mathrm{km}$ horizontally), higher than that of, for example, the ECMWF "ERA5 hourly data on pressure levels" product ($\approx 25\mathrm{km}$). The vertical resolution is of the same order here and in the above-mentionned ERA-5 product (49 and 37 vertical levels)." About using the native resolution of 137 levels, it may indeed increase the resolution in the boundary layer, but our main point (and issue) is the horizontal resolution.*

– L585: It is confusing that 7 degrees are used here whereas in the main text, 12 degrees was found to be the threshold to be used. Is this an analysis that preceded the analysis of the main text? It would be helpful then if this is mentioned in the text. *A descritpion of the subsections was added at the beginning of the section. "Section B1 describes the inversion methods and the differences with the one described in the main text. Section B2 and B3 are construct in the same model as Sections 5.1 and 5.2 with for the first subsection a preliminary analysis (independent of the decision tree) and for the*
*second the analysis of the decision tree method results."*

      – Fig. A1 is not referenced anywhere in the text. Please add a reference somewhere and put the figure at this place. *A reference to the annex has been added in section 2.4 . We have also extended the annex (and put a reference in it to figure A1) with a more mathematical description of the process as it was subject of numerous comments in all reviews.*

      – Fig. B1: Why is the GP2 line in panel b different from Fig. 3? They should be the same. If I understand it correctly, this
figure is the same as Fig. 3 in the main text. If not, please explain what is the difference. *In figure 3, the x-axis for the spatial varability of the wind direction is inverted (goes from 1 to 0), whereas it is in the classic direction in the annex. We choose to invert the direction to put into light the closeness between the two curves. A sentence has been added in figure 3 label to highlight this : "Note that the x-axis is plotted in the direction of decreasing spatial variability of wind direction (i.e. inverse axis) and increasing wind speed."*

– Fig. B1: The graphs inside each panel overlap with them quite significantly which makes them difficult to read. There must be another way to plot this, e.g. by adding another figure or decreasing the size of the small graphs and placing them better in each panel. *We decrease the size of the incrusted panels, replaced them and make the dotted lines behind them semi-transparent. We think that the information are now more readable.*

– Fig. B1: The y-axis of the small graphs does not have a label so that it is not clear what is illustrated there. It is also not described in the figure caption. *We extend the figure caption and hope that it will be sufficient.*

– Fig. B1: Panel c has an x-axis different from the other panels. Please explain and add the correct label for the x axis. *We extend the figure caption to deal with this.*

– Fig. B1: The caption does not describe what is actually shown in the figure. Please add this information or at least a reference to the similar figure in the main text. *New caption : "Sensitivity of the emission estimation error to different variables of interest. For each subfigure, the main panel shows the evolution of the error distribution as a function of the quantile of the variable of interest: the solid line indicates the median, the dotted lines the 1st and 3rd quartiles, the highlighted area the quantiles at 15.9% and 84.1%. The small incrusted panel shows the values taken by the variables of interest for the different quantiles. Subfigure (c) is an exception : as we have only one value of emission budget per city, we plot the evolution of the error distribution as a function of the rank of the city regarding the variable of interest. The optimized radius shown in panel (g) is a parameter of the gaussian plume models (see section 4.3) and is therefore not calculated for the other methods."*

– L603 and L606: Please remove "very" as this it is not clear what makes it "very" instead of just similar / different. *Correction made.*

– L612: same comment as above: Why 7 degrees? *see previous answer.*

**2 Technical corrections**

– L120: typo: approximately *Corrected*

– L144: typo Generation *Corrected*

– L170: remove dot between "selection" and "is" *Corrected*

– L175: remove dash between New and York *Corrected*

– L224: have –> has *Corrected*

– L314: sur –> such *Corrected*

– L360: Supp.Mat. –> Appendix *Corrected*

– L535: Annex –> Appendix *Corrected*

- L544: enabless –> enables *Corrected*

- L579: add a "to" between "referred" and "as": referred to as GP2 *Corrected*

**References**

Danjou, A., Broquet, G., Lian, J., Bréon, F.-M., and Lauvaux, T.: Evaluation of light atmospheric plume inversion methods using synthetic XCO2 satellite images to compute Paris CO2 emissions, Remote Sensing of Environment, 305, 113 900, https://doi.org/10.1016/j.rse.2023.113900, 2024.

Kuhlmann, G., Broquet, G., Marshall, J., Clément, V., Löscher, A., Meijer, Y., and Brunner, D.: Detectability of $CO_2$ emission plumes of cities and power plants with the Copernicus Anthropogenic $CO_2$ Monitoring (CO2M) mission, Atmospheric Measurement Techniques Discussions, pp. 1–35, https://doi.org/10.5194/amt-2019-180, 2019.

Worden, R. J., Doran, G., Kulawik, S., Eldering, A., Crisp, D., Frankenberg, C., O'Dell, C., and Bowman, K. W.: Evaluation and attribution of OCO-2 $XCO_2$ uncertainties, Atmospheric Measurement Techniques, 10, 2759–2771, https://doi.org/10.5194/amt-10-2759-2017, 2017.

---

## Author Response (AR3)

Dear M. Butz,

Thank you for your comment.

One of our co-authors, who is a native english speaker, re-read the article carefully with your comment in mind and made some corrections.

However those are pretty small : some typo (e.g "centre" on multiple places changed to "center", "syntethic" to "synthetic",...) and two slight corrections in sentences (on line 135 and 144).

If you have examples of other sentences that are not correct enough, could you point them to us?

Kind regards,

Alexandre Danjou